# Weighted Linear Bandits for Non-Stationary Environments

**Yoan Russac**
CNRS, Inria, ENS, Université PSL
`yoan.russac@ens.fr`

**Claire Vernade**
Deepmind
`vernade@google.com`

**Olivier Cappé**
CNRS, Inria, ENS, Université PSL
`olivier.cappe@cnrs.fr`

## Abstract

We consider a stochastic linear bandit model in which the available actions correspond to arbitrary context vectors whose associated rewards follow a *non-stationary* linear regression model. In this setting, the unknown regression parameter is allowed to vary in time. To address this problem, we propose `D-LinUCB`, a novel optimistic algorithm based on discounted linear regression, where exponential weights are used to smoothly forget the past. This involves studying the deviations of the sequential weighted least-squares estimator under generic assumptions. As a by-product, we obtain novel deviation results that can be used beyond non-stationary environments. We provide theoretical guarantees on the behavior of `D-LinUCB` in both slowly-varying and abruptly-changing environments. We obtain an upper bound on the dynamic regret that is of order $d^{2/3}B_T^{1/3}T^{2/3}$, where $B_T$ is a measure of non-stationarity ($d$ and $T$ being, respectively, dimension and horizon). This rate is known to be optimal. We also illustrate the empirical performance of `D-LinUCB` and compare it with recently proposed alternatives in simulated environments.

## 1 Introduction

Multi-armed bandits offer a class of models to address sequential learning tasks that involve exploration-exploitation trade-offs. In this work we are interested in structured bandit models, known as stochastic linear bandits, in which linear regression is used to predict rewards [1, 2, 22].

A typical application of bandit algorithms based on the linear model is online recommendation where actions are items to be, for instance, efficiently arranged on personalized web pages to maximize some conversion rate. However, it is unlikely that customers' preferences remain stable and the collected data becomes progressively obsolete as the interest for the items evolve. Hence, it is essential to design adaptive bandit agents rather than restarting the learning from scratch on a regular basis. In this work, we consider the use of weighted least-squares as an efficient method to progressively forget past interactions. Thus, we address sequential learning problems in which the parameter of the linear bandit is evolving with time.

Our first contribution consists in extending existing deviation inequalities to sequential weighted least-squares. Our result applies to a large variety of bandit problems and is of independent interest. In particular, it extends the recent analysis of heteroscedastic environments by [18]. It can also be useful to deal with class imbalance situations, or, as we focus on here, in non-stationary environments.

As a second major contribution, we apply our results to propose `D-LinUCB`, an adaptive linear bandit algorithm based on carefully designed exponential weights. `D-LinUCB` can be implemented fully recursively —without requiring the storage of past actions— with a numerical complexity that is comparable to that of `LinUCB`. To characterize the performance of the algorithm, we provide a unified regret analysis for abruptly-changing or slowly-varying environments.

The setting and notations are presented below and we state our main deviation result in Section 2. Section 3 is dedicated to non-stationary linear bandits: we describe our algorithms and provide regret upper bounds in abruptly-changing and slowly-varying environments. We complete this theoretical study with a set of experiments in Section 4.

## 1.1 Model and Notations

The setting we consider in this paper is a non-stationary variant of the stochastic linear bandit problem considered in [1, 22], where, at each round $t \geq 1$, the learner

- receives a finite set of feasible actions $\mathcal{A}_t \subset \mathbb{R}^d$;
- chooses an action $A_t \in \mathcal{A}_t$ and receives a reward $X_t$ such that

$$X_t = \langle A_t, \theta_t^\star \rangle + \eta_t, \tag{1}$$

  where $\theta_t^\star \in \mathbb{R}^d$ is an unknown parameter and $\eta_t$ is, conditionally on the past, a $\sigma-$subgaussian random noise.

The action set $\mathcal{A}_t$ may be arbitrary but its components are assumed to be bounded, in the sense that $\|a\|_2 \leq L, \forall a \in \mathcal{A}_t$. The time-varying parameter is also assumed to be bounded: $\forall t, \|\theta_t^\star\|_2 \leq S$. We further assume that $|\langle a, \theta_t^\star \rangle| \leq 1, \forall t, \forall a \in \mathcal{A}_t$, (obviously, this could be guaranteed by assuming that $L = S = 1$, but we indicate the dependence in $L$ and $S$ in order to facilitate the interpretation of some results). For a positive definite matrix $M$ and a vector $x$, we denote by $\|x\|_M$ the norm $\sqrt{x^\top M x}$.

The goal of the learner is to minimize the expected *dynamic regret* defined as

$$R(T) = \mathbb{E}\left[\sum_{t=1}^T \max_{a \in \mathcal{A}_t} \langle a, \theta_t^\star \rangle - X_t\right] = \sum_{t=1}^T \max_{a \in \mathcal{A}_t} \langle a - A_t, \theta_t^\star \rangle. \tag{2}$$

Even in the stationary case —i.e., when $\theta_t^\star = \theta^\star$—, there is, in general, no single fixed best action in this model.

When making stronger structural assumption on $\mathcal{A}_t$, one recovers specific instances that have also been studied in the literature. In particular, the canonical basis of $\mathbb{R}^d$, $\mathcal{A}_t = \{e_1, \ldots, e_d\}$, yields the familiar —non contextual— multi-armed bandit model [20]. Another variant, studied by [15] and others, is obtained when $\mathcal{A}_t = \{e_1 \otimes a_t, \ldots, e_k \otimes a_t\}$, where $\otimes$ denotes the Kronecker product and $a_t$ is a time-varying context vector shared by the $k$ actions.

## 1.2 Related Work

There is an important literature on online learning in changing environments. For the sake of conciseness, we restrict the discussion to works that consider specifically the stochastic linear bandit model in (1), including its restriction to the simpler (non-stationnary) multi-armed bandit model. Note that there is also a rich line of works that consider possibly non-linear contextual models in the case where one can make probabilistic assumptions on the contexts [10, 23].

Controlling the regret with respect to the non-stationary optimal action defined in (2) depends on the assumptions that are made on the time-variations of $\theta_t^\star$. A generic way of quantifying them is through a *variation bound* $B_T = \sum_{s=1}^{T-1} \|\theta_s^\star - \theta_{s+1}^\star\|_2$ [4, 6, 11], similar to the penalty used in the group fused Lasso [8]. The main advantage of using the variation budget is that is includes both *slowly-varying* and *abruptly-changing* environments. For the $K-$armed bandits with known $B_T$, [4–6] achieve the tight dynamic regret bound of $O(K^{1/3}B_T^{1/3}T^{2/3})$. For linear bandits, [11, 12] propose an algorithm based on the use of a sliding-window and provide a $O(d^{2/3}B_T^{1/3}T^{2/3})$ dynamic regret bound; since this contribution is close to ours, we discuss it further in Section 3.2.

A more specific non-stationary setting arises when the number of changes in the parameter is bounded by $\Gamma_T$, as in traditional change-point models. The problem is usually referred to as *switching bandits* or *abruptly-changing* environments. It is, for instance, the setting considered in the work by Garivier and Moulines [14], who analyzed the dynamic regret of UCB strategies based on either a sliding-window or exponential discounting. For both policies, they prove upper bounds on the regret in $O(\sqrt{\Gamma_T T})$ when $\Gamma_T$ is known. They also provide a lower bound in a specific non-stationary setting, showing that $R(T) = \Omega(\sqrt{T})$. The algorithm ideas can be traced back to [19]. [28] shows that an horizon-independent version of the sliding window algorithm can also be analyzed in a slowly-varying setting. [17] analyze windowing and discounting approaches to address dynamic pricing guided by a (time-varying) linear regression model. Discount factors have also been used with Thomson sampling in dynamic environments as in [16, 26].

In abruptly-changing environments, the alternative approach relies on change-point detection [3, 7, 9, 29, 30]. A bound on the regret in $O((\frac{1}{\epsilon^2} + \frac{1}{\Delta}) \log(T))$ is proven by [30], where $\epsilon$ is the smallest gap that can be detected by the algorithm, which had to be given as prior knowledge. [9] proves a minimax bound in $O(\sqrt{\Gamma_T K T})$ if $\Gamma_T$ is known. [7] achieves a rate of $O(\sqrt{\Gamma_T K T})$ without any prior knowledge of the gaps or $\Gamma_T$. In the contextual case, [29] builds on the same idea: they use a pool of `LinUCB` learners called *slave models* as experts and they add a new model when no existing slave is able to give good prediction, that is, when a change is detected. A limitation however of such an approach is that it can not adapt to some slowly-varying environments, as will be illustrated in Section 4. From a practical viewpoint, the methods based either on sliding window or change-point detection require the storage of past actions whereas those based on discount factors can be implemented fully recursively.

Finally, non-stationarity may also arise in more specific scenarios connected, for instance, to the decaying attention of the users, as investigated in [21, 24, 27]. In the following, we consider the general case where the parameters satisfy the variation bound, i.e., $\sum_{t=1}^{T-1} \|\theta_t^\star - \theta_{t+1}^\star\|_2 \leq B_T$ and we propose an algorithm based on discounted linear regression.

## 2   Confidence Bounds for Weighted Linear Bandits

In this section, we consider the concentration of the weighted regularized least-squares estimator, when used with general weights and regularization parameters. To the best of our knowledge there is no such results in the literature for sequential learning —i.e., when the current regressor may depend on the random outcomes observed in the past. The particular case considered in Lemma 5 of [18] (heteroscedastic noise with optimal weights) stays very close to the unweighted case and we show below how to extend this result. We believe that this new bound is of interest beyond the specific model considered in this paper. For the sake of clarity, we first focus on the case of regression models with fixed parameter, where $\theta_t^\star = \theta^\star$, for all $t$.

First consider a deterministic sequence of regularization parameters $(\lambda_t)_{t \geq 1}$. The reason why these should be non-constant for weighted least-squares will appear clearly in Section 3. Next, define by $\mathcal{F}_t = \sigma(X_1, \dots, X_t)$ the filtration associated with the random observations. We assume that both the actions $A_t$ and positive weights $w_t$ are predictable, that is, they are $\mathcal{F}_{t-1}$ measurable.

Defining by

$$\hat{\theta}_t = \arg\min_{\theta \in \mathbb{R}^d} \left( \sum_{s=1}^t w_s (X_s - \langle A_s, \theta \rangle)^2 + \lambda_t \|\theta\|_2^2 \right)$$

the regularized weighted least-squares estimator of $\theta^\star$ at time $t$, one has

$$\hat{\theta}_t = V_t^{-1} \sum_{s=1}^t w_s A_s X_s \quad \text{where} \quad V_t = \sum_{s=1}^t w_s A_s A_s^\top + \lambda_t I_d, \tag{3}$$

and $I_d$ denotes the $d$-dimensional identity matrix. We further consider an arbitrary sequence of positive parameters $(\mu_t)_{t \geq 1}$ and define the matrix

$$\widetilde{V}_t = \sum_{s=1}^t w_s^2 A_s A_s^\top + \mu_t I_d. \tag{4}$$

$\widetilde{V}$ is strongly connected to the variance of the estimator $\hat{\theta}_t$, which involves the squares of the weights $(w_s^2)_{s \geq 1}$. For the time being, $\mu_t$ is arbitrary and will be set as a function of $\lambda_t$ in order to optimize the deviation inequality.

We then have the following maximal deviation inequality.

**Theorem 1.** *For any $\mathcal{F}_t$-predictable sequences of actions $(A_t)_{t \geq 1}$ and positive weights $(w_t)_{t \geq 1}$ and for all $\delta > 0$,*

$$\mathbb{P}\left(\forall t, \|\hat{\theta}_t - \theta^\star\|_{V_t \widetilde{V}_t^{-1} V_t} \leq \frac{\lambda_t}{\sqrt{\mu_t}} S + \sigma \sqrt{2 \log(1/\delta) + d \log\left(1 + \frac{L^2 \sum_{s=1}^t w_s^2}{d \mu_t}\right)}\right) \geq 1 - \delta.$$

The proof of this theorem is deferred to the appendix and combines an argument using the method of mixtures and the use of a proper stopping time. The standard result used for least-squares [20, Chapter 20] is recovered by taking $\mu_t = \lambda_t$ and $w_t = 1$ (note that $\widetilde{V}_t$ is then equal to $V_t$). When the weights are not equal to 1, the appearance of the matrix $\widetilde{V}_t$ is a consequence of the fact that the variance terms are proportional to the squared weights $w_t^2$, while the least-squares estimator itself is defined with the weights $w_t$. In the weighted case, the matrix $V_t \widetilde{V}_t^{-1} V_t$ must be used to define the confidence ellipsoid.

An important property of the least-squares estimator is to be scale-invariant, in the sense that multiplying all weights $(w_s)_{1 \leq s \leq t}$ and the regularization parameter $\lambda_t$ by a constant leaves the estimator $\hat{\theta}_t$ unchanged. In Theorem 1, the only choice of sequence $(\mu_t)_{t \geq 1}$ that is compatible with this scale-invariance property is to take $\mu_t$ proportional to $\lambda_t^2$: then the matrix $V_t \widetilde{V}_t^{-1} V_t$ becomes scale-invariant (*i.e.* unchanged by the transformation $w_s \mapsto \alpha w_s$) and so does the upper bound of $\|\hat{\theta}_t - \theta^\star\|_{V_t \widetilde{V}_t^{-1} V_t}$ in Theorem 1. In the following, we will stick to this choice, while particularizing the choice of the weights $w_t$ to allow for non-stationary models.

It is possible to extend this result to heteroscedastic noise, when $\eta_t$ is $\sigma_t$ sub-Gaussian and $\sigma_t$ is $\mathcal{F}_{t-1}$ measurable, by defining $\widetilde{V}_t$ as $\sum_{s=1}^t w_s^2 \sigma_s^2 A_s A_s^\top + \mu_t I_d$. In the next section, we will also use an extension of Theorem 1 to the non-stationary model presented in (1) . In this case, Theorem 1 holds with $\theta^\star$ replaced by $V_t^{-1}\left(\sum_{s=1}^t w_s A_s A_s^\top \theta_s^\star + \lambda_t \theta_r^\star\right)$, where $r$ is an arbitrary time index (proposition 3 in Appendix). The fact that $r$ can be chosen freely is a consequence of the assumption that the sequence of L2-norms of the parameters $(\theta_t^\star)_{t \geq 1}$ is bounded by $S$.

## 3 Application to Non-stationary Linear Bandits

In this section, we consider the non-stationary model defined in (1) and propose a bandit algorithm in Section 3.1, called Discounted Linear Upper Confidence Bound (D-LinUCB), that relies on weighted least-squares to adapt to changes in the parameters $\theta_t^\star$. Analyzing the performance of D-LinUCB in Section 3.2, we show that it achieves reliable performance both for abruptly changing or slowly drifting parameters.

### 3.1 The D-LinUCB Algorithm

Being adaptive to parameter changes indeed implies to reduce the influence of observations that are far back in the past, which suggests using weights $w_t$ that increase with time. In doing so, there are two important caveats to consider. First, this can only be effective if the sequence of weights is growing sufficiently fast (see the analysis in the next section). We thus consider exponentially increasing weights of the form $w_t = \gamma^{-t}$, where $0 < \gamma < 1$ is the discount factor.

Next, due to the absence of assumptions on the action sets $\mathcal{A}_t$, the regularization is instrumental in obtaining guarantees of the form given in Theorem 1. In fact, if $w_t = \gamma^{-t}$ while $\lambda_t$ does not increase sufficiently fast, then the term $\log\left(1 + (L^2 \sum_{s=1}^t w_s^2)/(d \mu_t)\right)$ will eventually dominate the radius of the confidence region since we choose $\mu_t$ proportional to $\lambda_t^2$. This occurs because there is no guarantee that the algorithm will persistently select actions $A_t$ that span the entire space. With this in mind, we consider an increasing regularization factor of the form $\lambda_t = \gamma^{-t} \lambda$, where $\lambda > 0$ is a hyperparameter.

Note that due to the scale-invariance property of the weighted least-square estimator, we can equivalently consider that at time $t$, we are given *time-dependent* weights $w_{t,s} = \gamma^{t-s}$, for $1 \le s \le t$ and that $\hat{\theta}_t$ is defined as

$$\underset{\theta \in \mathbb{R}^d}{\arg\min} \Big( \sum_{s=1}^{t} \gamma^{t-s} (X_s - \langle A_s, \theta \rangle)^2 + \lambda \|\theta\|_2^2 \Big).$$

For numerical stability reasons, this form is preferable and is used in the statement of Algorithm 1. In the analysis of Section 3.2 however we revert to the standard form of the weights, which is required to apply the concentration result of Section 1. We are now ready to describe D-LinUCB in Algorithm 1.

---

**Algorithm 1:** D-LinUCB

---

**Input:** Probability $\delta$, subgaussianity constant $\sigma$, dimension $d$, regularization $\lambda$, upper bound for actions $L$, upper bound for parameters $S$, discount factor $\gamma$.
**Initialization:** $b = 0_{\mathbb{R}^d}$, $V = \lambda I_d$, $\widetilde{V} = \lambda I_d$, $\hat{\theta} = 0_{\mathbb{R}^d}$
**for** $t \ge 1$ **do**

    Receive $\mathcal{A}_t$, compute $\beta_{t-1} = \sqrt{\lambda} S + \sigma \sqrt{2 \log\left(\frac{1}{\delta}\right) + d \log\left(1 + \frac{L^2(1-\gamma^{2(t-1)})}{\lambda d(1-\gamma^2)}\right)}$

    **for** $a \in \mathcal{A}_t$ **do**
        Compute $\text{UCB}(a) = a^\top \hat{\theta} + \beta_{t-1} \sqrt{a^\top V^{-1} \widetilde{V} V^{-1} a}$
    $A_t = \arg\max_a (\text{UCB}(a))$
    **Play action** $A_t$ and **receive reward** $X_t$
    **Updating phase**: $V = \gamma V + A_t A_t^\top + (1-\gamma)\lambda I_d$, $\widetilde{V} = \gamma^2 \widetilde{V} + A_t A_t^\top + (1-\gamma^2)\lambda I_d$
        $b = \gamma b + X_t A_t$, $\hat{\theta} = V^{-1} b$

---

## 3.2 Analysis

As discussed previously, we consider weights of the form $w_t = \gamma^{-t}$ (where $0 < \gamma < 1$) in the D-LinUCB algorithm. In accordance with the discussion at the end of Section 1, Algorithm 1 uses $\mu_t = \gamma^{-2t}\lambda$ as the parameter to define the confidence ellipsoid around $\hat{\theta}_{t-1}$. The confidence ellipsoid $\mathcal{C}_t$ is defined as $\left\{ \theta : \|\theta - \hat{\theta}_{t-1}\|_{V_{t-1}\widetilde{V}_{t-1}^{-1}V_{t-1}} \le \beta_{t-1} \right\}$ where

$$\beta_t = \sqrt{\lambda} S + \sigma \sqrt{2 \log(1/\delta) + d \log\left(1 + \frac{L^2(1-\gamma^{2t})}{\lambda d(1-\gamma^2)}\right)}. \tag{5}$$

Using standard algebraic calculations together with the remark above about scale-invariance it is easily checked that at time $t$ Algorithm 1 selects the action $A_t$ that maximizes $\langle a, \theta \rangle$ for $a \in \mathcal{A}_t$ and $\theta \in \mathcal{C}_t$. The following theorem bounds the regret resulting from Algorithm 1.

**Theorem 2.** *Assuming that $\sum_{s=1}^{T-1} \|\theta_s^\star - \theta_{s+1}^\star\|_2 \le B_T$, the regret of the D-LinUCB algorithm is bounded for all $\gamma \in (0,1)$ and integer $D \ge 1$, with probability at least $1-\delta$, by*

$$R_T \le 2LDB_T + \frac{4L^3 S}{\lambda} \frac{\gamma^D}{1-\gamma} T + 2\sqrt{2}\beta_T \sqrt{dT} \sqrt{T \log(1/\gamma) + \log\left(1 + \frac{L^2}{d\lambda(1-\gamma)}\right)}. \tag{6}$$

The first two terms of the r.h.s. of (6) are the result of the bias due to the non-stationary environment. The last term is the consequence of the high probability bound established in the previous section and an adaptation of the technique used in [1].

We give the complete proof of this result in appendix. The high-level idea of the proof is to isolate bias and variance terms. However, in contrast with the stationary case, the confidence ellipsoid $\mathcal{C}_t$ does not necessarily contain (with high probability) the actual parameter value $\theta_t^\star$ due to the (unknown) bias arising from the time variations of the parameter. We thus define

$$\bar{\theta}_t = V_{t-1}^{-1} \left( \sum_{s=1}^{t-1} \gamma^{-s} A_s A_s^\top \theta_s^\star + \lambda \gamma^{-(t-1)} \theta_t^\star \right)$$

which is an action-dependent analogue of the parameter value $\theta^\star$ in the stationary setting (although this is a random value). As mentioned in section 2, $\bar{\theta}_t$ does belong to $\mathcal{C}_t$ with probability at least $1 - \delta$ (see Proposition 3 in Appendix). The regret may then be split as

$$R_T \leq 2L \sum_{t=1}^{T} \|\theta_t^\star - \bar{\theta}_t\|_2 + \sum_{t=1}^{T} \langle A_t, \theta_t - \bar{\theta}_t \rangle \quad \text{(with probability at least } 1 - \delta),$$

where $(A_t, \theta_t) = \arg\max_{(a \in \mathcal{A}_t, \theta \in \mathcal{C}_t)} \langle a, \theta \rangle$. The rightmost term can be handled by proceeding as in the case of stationary linear bandits, thanks to the deviation inequality obtained in Section 2. The first term in the r.h.s. can be bounded deterministically, from the assumption made on $\sum_{s=1}^{T-1} \|\theta_s^\star - \theta_{s+1}^\star\|_2$. In doing so, we introduce the analysis parameter $D$ that, roughly speaking, corresponds to the window length equivalent to a particular choice of discount factor $\gamma$: the bias resulting from observations that are less than $D$ time steps apart may be bounded in term of $D$ while the remaining ones are bounded globally by the second term of the r.h.s. of (6). This sketch of proof is substantially different from the arguments used by [11] to analyze their sliding window algorithm (called `SW-LinUCB`). We refer to the appendix for a more detailed analysis of these differences. Interestingly, the regret bound of Theorem 2 holds despite the fact that the true parameter $\theta_t^\star$ may not be contained in the confidence ellipsoid $\mathcal{C}_{t-1}$, in contrast to the proof of [14].

It can be checked that, as $T$ tends to infinity, the optimal choice of the analysis parameter $D$ is to take $D = \log(T)/(1 - \gamma)$. Further assuming that one may tune $\gamma$ as a function of the horizon $T$ and the variation upper bound $B_T$ yields the following result.

**Corollary 1.** *By choosing $\gamma = 1 - (B_T/(dT))^{2/3}$, the regret of the `D-LinUCB` algorithm is asymptotically upper bounded with high probability by a term $O(d^{2/3} B_T^{1/3} T^{2/3})$ when $T \to \infty$.*

This result is favorable as it corresponds to the same order as the lower bound established by [4]. More precisely, the case investigated by [4] corresponds to a non-contextual model with a number of changes that grows with the horizon. On the other hand, the guarantee of Corollary 1 requires horizon-dependent tuning of the discount factor $\gamma$, which opens interesting research issues (see also [11]).

## 4   Experiments

This section is devoted to the evaluation of the empirical performance of `D-LinUCB`. We first consider two simulated low-dimensional environments that illustrate the behavior of the algorithms when confronted to either abrupt changes or slow variations of the parameters. The analysis of the previous section, suggests that `D-LinUCB` should behave properly in both situations. We then consider a more realistic scenario in Section 4.2, where the contexts are high-dimensional and extracted from a data set of actual user interactions with a web service.

For benchmarking purposes, we compare `D-LinUCB` to the Dynamic Linear Upper Confidence Bound (`dLinUCB`) algorithm proposed by [29] and with the Sliding Window Linear UCB (`SW-LinUCB`) of [11]. The principle of the `dLinUCB` algorithm is that a master bandit algorithm is in charge of choosing the best `LinUCB` slave bandit for making the recommendation. Each slave model is built to run in each one of the different environments. The choice of the slave model is based on a lower confidence bound for the so-called *badness* of the different models. The badness is defined as the number of times the expected reward was found to be far enough from the actual observed reward on the last $\tau$ steps, where $\tau$ is a parameter of the algorithm. When a slave is chosen, the action proposed to a user is the result of the `LinUCB` algorithm associated with this slave. When the action is made, all the slave models that were good enough are updated and the models whose badness were too high are deleted from the pool of slaves models. If none of the slaves were found to be sufficiently good, a new slave is added to the pool.

The other algorithm that we use for comparison is `SW-LinUCB`, as presented in [11]. Rather than using exponentially increasing weights, a hard threshold is adopted. Indeed, the actions and rewards included in the $l$-length sliding window are used to estimate the linear regression coefficients. We expect `D-LinUCB` and `SW-LinUCB` to behave similarly as they both may be shown to have the same sort of regret guarantees (see appendix).

In the case of abrupt changes, we also compare these algorithms to the Oracle Restart LinUCB (`LinUCB-OR`) strategy that would know the change-points and simply restart, after each change, a

new instance of the `LinUCB` algorithm. The regret of this strategy may be seen as an empirical lower bound on the optimal behavior of an online learning algorithm in abruptly changing environments.

In the following figures, the vertical red dashed lines correspond to the change-points (in abrupt changes scenarios). They are represented to ease the understanding but except for `LinUCB-OR`, they are of course unknown to the learning algorithms. When applicable, the blue dashed lines correspond to the average detection time of the breakpoints with the `dLinUCB` algorithm. For `D-LinUCB` the discount parameter is chosen as $\gamma = 1 - (\frac{B_T}{dT})^{2/3}$. For `SW-LinUCB` the window's length is set to $l = (\frac{dT}{B_T})^{2/3}$, where $d = 2$ in the experiment. Those values are theoretically supposed to minimize the asymptotic regret. For the Dynamic Linear UCB algorithm, the badness is estimated from $\tau = 200$ steps, as in the experimental section of [29].

## 4.1 Synthetic data in abruptly-changing or slowly-varying scenarios

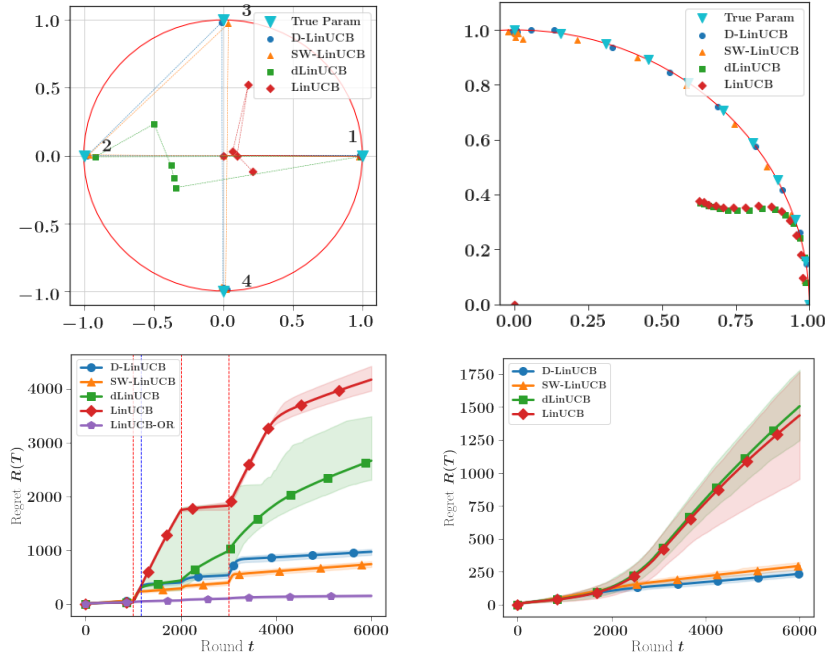

Figure 1: Performances of the algorithms in the abruptly-changing environment (on the left), and, the slowly-varying environment (on the right). The upper plots correspond to the estimated parameter and the lower ones to the accumulated regret, both are averaged on $N = 100$ independent experiments

In this first experiment, we observe the empirical performance of all algorithms in an abruptly changing environment of dimension 2 with 3 breakpoints. The number of rounds is set to $T = 6000$. The light blue triangles correspond to the different positions of the true unknown parameter $\theta_t^\star$: before $t = 1000$, $\theta_t^\star = (1, 0)$; for $t \in [\![1000, 2000]\!]$, $\theta_t^\star = (-1, 0)$; for $t \in [\![2000, 3000]\!]$, $\theta_t^\star = (0, 1)$; and, finally, for $t > 3000$, $\theta_t^\star = (0, -1)$. This corresponds to a hard problem as the sequence of parameters is widely spread in the unit ball. Indeed it forces the algorithm to adapt to big changes, which typically requires a longer adaptation phase. On the other hand, it makes the detection of changes easier, which is an advantage for `dLinUCB`. In the second half of the experiment (when $t \geq 3000$) there is no change, `LinUCB` struggles to catch up and suffers linear regret for long periods after the last change-point. The results of our simulations are shown in the left column of Figure 1. On the top row we show a 2-dimensional scatter plot of the estimate of the unknown parameters $\hat{\theta}_t$ every 1000 steps averaged on 100 independent experiment. The bottom row corresponds to the regret averaged over 100 independent experiments with the upper and the lower 5% quantiles. In this environment, with 1-subgaussian random noise, `dLinUCB` struggles to detect the change-points. Over the 100 experiments, the first change-point was detected in 95% of the runs, the second was never detected and the third only in 6% of the runs, thus limiting the effectiveness of the `dLinUCB` approach. When decreasing the variance of the noise, the performance of `dLinUCB` improves and gets closer to

the performance of the oracle restart strategy `LinUCB-OR`. It is worth noting that for both `SW-LinUCB` and `D-LinUCB`, the estimator $\hat{\theta}_t$ adapts itself to non-stationarity and is able to follow $\theta_t^\star$ (with some delay), as shown on the scatter plot. Predictably, `LinUCB-OR` achieves the best performance by restarting exactly whenever a change-point happens.

The second experiment corresponds to a slowly-changing environment. It is easier for `LinUCB` to keep up with the adaptive policies in this scenario. Here, the parameter $\theta_t^\star$ starts at $(1$ and moves continuously counter-clockwise on the unit-circle up to the position $[0, 1]$ in 3000 steps. We then have a steady period of 3000 steps. For this sequence of parameters, $B_T = \sum_{t=1}^{T-1} \|\theta_t^\star - \theta_{t+1}^\star\|_2 = 1.57$. The results are reported in the right column of Figure 1. Unsurprisingly, `dLinUCB` does not detect any change and thus displays the same performance as `LinUCB`. `SW-LinUCB` and `D-LinUCB` behaves similarly and are both robust to such an evolution in the regression parameters. The performance of `LinUCB-OR` is not reported here, as restarting becomes ineffective when the changes are too frequent (here, during the first 3000 time steps, there is a change at every single step). The scatter plot also gives interesting information: $\hat{\theta}_t$ tracks $\theta_t^\star$ quite effectively for both `SW-LinUCB` and `D-LinUCB` but the two others algorithms lag behind. `LinUCB` will eventually catch up if the length of the stationary period becomes larger.

## 4.2 Simulation based on a real dataset

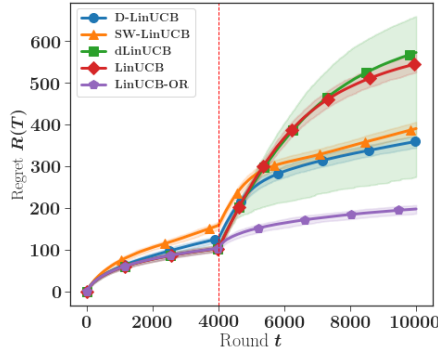

Figure 2: Behavior of the different algorithms on large-dimensional data

`D-LinUCB` also performs well in high-dimensional space ($d = 50$). For this experiment, a dataset providing a sample of 30 days of Criteo live traffic data [13] was used. It contains banners that were displayed to different users and contextual variables, including the information of whether the banner was clicked or not. We kept the categorical variables $cat1$ to $cat9$, together with the variable $campaign$, which is a unique identifier of each campaign. Beforehand, these contexts have been one-hot encoded and 50 of the resulting features have been selected using a Singular Value Decomposition. $\theta^\star$ is obtained by linear regression. The rewards are then simulated using the regression model with an additional Gaussian noise of variance $\sigma^2 = 0.15$. At each time step, the different algorithms have the choice between two 50-dimensional contexts drawn at random from two separate pools of 10000 contexts corresponding, respectively, to clicked or not clicked banners. The non-stationarity is created by switching 60% of $\theta^\star$ coordinates to $-\theta^\star$ at time 4000, corresponding to a partial class inversion. The cumulative dynamic regret is then averaged over 100 independent replications. The results are shown on Figure 2. In the first stationary period, `LinUCB` and `dLinUCB` perform better than the adaptive policies by using all available data, whereas the adaptive policies only use the most recent events. After the breakpoint, `LinUCB` suffers a large regret, as the algorithm fails to adapt to the new environment. In this experiment, `dLinUCB` does not detect the change-point systematically and performs similarly as `LinUCB` on average, it can still outperform adaptive policies from time to time when the breakpoint is detected as can be seen with the 5% quantile. `D-LinUCB` and `SW-LinUCB` adapt more quickly to the change-point and perform significantly better than the non-adaptive policies after the breakpoint. Of course, the oracle policy `LinUCB-OR` is the best performing policy. The take-away message is that there is no free lunch: in a stationary period by using only the most recent events `SW-LinUCB` and `D-LinUCB` do not perform as good as a policy that uses all the available information. Nevertheless, after a breakpoint, the recovery is much faster with the adaptive policies.

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
