[Supplementary Material · supplementary_NIPS.pdf]

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

# Appendix

## A  Confidence Bounds for Weighted Linear Bandits

### A.1  Preliminary results

In this section we give the main results for obtaining Theorem 1. For the sake of conciseness all the results will be stated with $\sigma$-subgaussian noises but the proofs will be done with the particular value of $\sigma = 1$. The model we consider is the one defined by equation (1), where we recall that $(\eta_s)_s$ is, conditionally on the past, a sequence of $\sigma$-subgaussian random noises. The results of this section are close to the one proposed in [1] but our results are valid with a sequence of predictable weights.

We introduce the quantity $S_t = \sum_{s=1}^{t} w_s A_s \eta_s$ and $\widetilde{V}_t = \sum_{s=1}^{t} w_s^2 A_s A_s^\top + \mu_t I_d$. When the regularization term is omitted, let $\widetilde{V}_t(0) = \sum_{s=1}^{t} w_s^2 A_s A_s^\top$. The filtration associated with the random observations is denoted $\mathcal{F}_t = \sigma(X_1, \ldots, X_t)$ such that $A_t$ is $\mathcal{F}_{t-1}$-measurable and $\eta_t$ is $\mathcal{F}_t$-measurable. The weights are also assumed to be predictable. The following lemma is an extension to the weighted case of Lemma 8 of [1].

**Lemma 1.** *Let $(w_t)_{t \geq 1}$ be a sequence of predictable and positive weights. Let $x \in \mathbb{R}^d$ be arbitrary and consider for any $t \geq 1$*

$$M_t(x) = \exp\left( \frac{1}{\sigma} x^\top S_t - \frac{1}{2} x^\top \widetilde{V}_t(0) x \right).$$

*Let $\tau$ be a stopping time with respect to the filtration $\{\mathcal{F}_t\}_{t=0}^{\infty}$. Then $M_\tau(x)$ is almost surely well-defined and*

$$\forall x \in \mathbb{R}^d, \mathbb{E}[M_\tau(x)] \leq 1.$$

*Proof.* First, we prove that $\forall x \in \mathbb{R}^d, (M_t(x))_{t=0}^{\infty}$ is a super-martingale.

Let $x \in \mathbb{R}^d$,

$$
\begin{aligned}
\mathbb{E}[M_t(x)|\mathcal{F}_{t-1}] &= \mathbb{E}\left[ \exp\left( x^\top S_{t-1} + x^\top w_t A_t \eta_t - 1/2 x^\top (\widetilde{V}_{t-1}(0) + w_t^2 A_t A_t^\top) x \right) | \mathcal{F}_{t-1} \right] \\
&= M_{t-1}(x) \mathbb{E}\left[ \exp(x^\top w_t A_t \eta_t - \frac{1}{2} w_t^2 x^\top A_t A_t^\top x) | \mathcal{F}_{t-1} \right] \\
&= M_{t-1}(x) \exp(-\frac{1}{2} w_t^2 x^\top A_t A_t^\top x) \mathbb{E}\left[ \exp(x^\top w_t A_t \eta_t) | \mathcal{F}_{t-1} \right] \\
&\leq M_{t-1}(x) \exp(-\frac{1}{2} w_t^2 x^\top A_t A_t^\top x) \exp(1/2 w_t^2 (x^\top A_t)^2) \\
&= M_{t-1}(x).
\end{aligned}
$$

The second equality comes from the fact that $S_{t-1}$ and $\widetilde{V}_{t-1}$ are $\mathcal{F}_{t-1}$-measurable. The inequality is the definition of the conditional 1-subgaussianity where we also use the $\mathcal{F}_{t-1}$-measurability of $w_t$.

Using this supermartingale property, we have $\mathbb{E}[M_t(x)] \leq 1$. The convergence theorem for non-negative supermartingales ensures that $M_\infty(x) = \lim_{t \to \infty} M_t(x)$ is almost surely well defined. By introducing the stopped supermartingale $\mathcal{M}_t(x) = M_{\min(t,\tau)}(x)$, we have $M_\tau(x) = \lim_{t \to \infty} \mathcal{M}_t(x)$. Knowing that $\mathcal{M}_t(x)$ is also a supermartingale, we have

$$\mathbb{E}[\mathcal{M}_t(x)] = \mathbb{E}[M_{\min(t,\tau)}(x)] \leq \mathbb{E}[M_{\min(0,\tau)}(x)] = \mathbb{E}[M_0(x)] = 1.$$

By using Fatou's lemma:

$$\mathbb{E}[M_\tau(x)] = \mathbb{E}[\liminf_{t \to \infty} \mathcal{M}_t(x)] \leq \liminf_{t \to \infty} \mathbb{E}[\mathcal{M}_t(x)] \leq 1.$$

$\square$

In the next lemma, we will integrate $M_t(x)$ with respect to a time-dependent probability measure. This is the key for allowing sequential regularizations in the concentration inequality stated in Theorem 1. This lemma is inspired by the method of mixtures first presented in [25]. The idea of

using time-varying probability measures is inspired from the proof of Theorem 11 in [18]. The two following lemmas are included in the appendix so that the article is self-contained. There are not a mere consequence of the results in [1] because of the time-dependent regularization parameters. As explained in Section 3, this is unavoidable when using exponential weights to avoid the vanishing effect of the regularization.

**Lemma 2.** *Let $(h_t)_t$ be a sequence of probability measures on $\mathbb{R}^d$. We define $\widetilde{M}_t = \int_{\mathbb{R}^d} M_t(x)dh_t(x)$. Then,*

$$\forall t, \mathbb{E}[\widetilde{M}_t] \leq 1$$

*Proof.*

$$\mathbb{E}[\widetilde{M}_t] = \int \widetilde{M}_t \, d\mathbb{P} = \int \left( \int_{\mathbb{R}^d} M_t(x)dh_t(x) \right) d\mathbb{P}$$

$$= \int_{\mathbb{R}^d} \left( \int M_t(x)d\mathbb{P} \right) dh_t(x) \quad \text{(Fubini's theorem)}$$

$$= \int_{\mathbb{R}^d} \mathbb{E}[M_t(x)]dh_t(x)$$

$$\leq \int_{\mathbb{R}^d} dh_t(x) \quad \text{(Lemma 1)}$$

$$\leq 1. \quad (h_t \text{ probability measure.})$$

$\square$

Lemma 2 is a warm-up for the next lemma and is helpful for understanding why Lemma 3 holds. It is valid for any fixed time $t$. The next step is to give its equivalent in a stopped version in the specific case of gaussian random vectors.

**Lemma 3.** *Let $(\mu_t)_t$ be a deterministic sequence of regularization parameters. Let $\mathcal{F}_\infty = \sigma\left(\cup_{t=1}^\infty \mathcal{F}_t\right)$ be the tail $\sigma$-algebra of the filtration $(\mathcal{F}_t)_t$. Let $X = (X_t)_{t \geq 1}$ be an independent sequence of gaussian random vectors such that $X_t \sim \mathcal{N}(0, \frac{1}{\mu_t}I_d) = h_t$ with $X$ independent of $\mathcal{F}_\infty$. We define*

$$\bar{M}_t(\mu_t) = \mathbb{E}[M_t(X_t)|\mathcal{F}_\infty] = \int_{\mathbb{R}^d} M_t(x)f_{\mu_t}(x)dx,$$

*where $f_{\mu_t}$ is the probability density function associated with $h_t$ defined as,*

$$f_{\mu_t}(x) = \frac{1}{\sqrt{(2\pi)^d \det(1/\mu_t I_d)}} \exp(-\frac{\mu_t x^\top x}{2}).$$

*Let $\tau$ be a stopping time with respect to the filtration $(\mathcal{F}_t)_t$ then,*

$$\mathbb{E}[\bar{M}_\tau(\mu_\tau)] \leq 1.$$

*Proof.* We can use the result of Lemma 1 which gives $\forall x \in \mathbb{R}^d$, $\mathbb{E}[M_\tau(x)] \leq 1$.

We have,

$$\mathbb{E}[\bar{M}_\tau(\mu_\tau)] = \mathbb{E}[\mathbb{E}[M_\tau(X_\tau)|\mathcal{F}_\infty]] = \mathbb{E}[\mathbb{E}[\mathbb{E}[M_\tau(X_\tau)|\mathcal{F}_\infty]|(X_t)_{t \geq 1}]]$$

$$= \mathbb{E}[\mathbb{E}[\mathbb{E}[M_\tau(X_\tau)|(X_t)_{t \geq 1}]|\mathcal{F}_\infty]] \leq 1.$$

The inequality is a consequence of Lemma 1 as, conditionally to the sequence $(X_t)_t$, $M_\tau(X_\tau)$ is of the form $M_\tau(x)$ with a fixed $x$. $\square$

We finally state the main result needed to obtain Theorem 1.

**Proposition 1.** *For $(w_s)_{s \geq 1}$ a sequence of predictable and positive weights, $\forall \delta > 0$, the following deviation inequality holds*

$$\mathbb{P}\left( \exists t \geq 0, \|S_t\|_{\widetilde{V}_t^{-1}} \geq \sigma \sqrt{2\log\left(\frac{1}{\delta}\right) + \log\left(\frac{\det(\widetilde{V}_t)}{\mu_t^d}\right)} \right) \leq \delta.$$

*Proof.* For a fixed $t$,

$$\bar{M}_t(\mu_t) = \int_{\mathbb{R}^d} M_t(x) f_{\mu_t}(x) dx$$

$$= \frac{1}{\sqrt{(2\pi)^d \det(1/\mu_t I_d)}} \int_{\mathbb{R}^d} \exp\left( x^\top S_t - \frac{1}{2}\|x\|^2_{\mu_t I_d} - \frac{1}{2}\|x\|^2_{\widetilde{V}_t(0)} \right) dx$$

$$= \frac{1}{\sqrt{(2\pi)^d \det(1/\mu_t I_d)}} \int_{\mathbb{R}^d} \exp\left( x^\top S_t - \frac{1}{2}\|x\|^2_{\widetilde{V}_t} \right) dx$$

$$= \frac{1}{\sqrt{(2\pi)^d \det(1/\mu_t I_d)}} \int_{\mathbb{R}^d} \exp\left( \frac{1}{2}\|S_t\|^2_{\widetilde{V}_t^{-1}} - \frac{1}{2}\|x - \widetilde{V}_t^{-1} S_t\|^2_{\widetilde{V}_t} \right) dx$$

$$= \frac{\exp\left( \frac{1}{2}\|S_t\|^2_{\widetilde{V}_t^{-1}} \right)}{\sqrt{(2\pi)^d \det(1/\mu_t I_d)}} \int_{\mathbb{R}^d} \exp\left( -\frac{1}{2}\|x - \widetilde{V}_t^{-1} S_t\|^2_{\widetilde{V}_t} \right) dx$$

$$= \frac{\exp\left( \frac{1}{2}\|S_t\|^2_{\widetilde{V}_t^{-1}} \right)}{\sqrt{(2\pi)^d \det(1/\mu_t I_d)}} \sqrt{(2\pi)^d \det\left( \widetilde{V}_t^{-1} \right)}$$

$$= \exp\left( \frac{1}{2}\|S_t\|^2_{\widetilde{V}_t^{-1}} \right) \sqrt{\frac{\det(\mu_t I_d)}{\det(\widetilde{V}_t)}}.$$

We introduce the particular stopping time,

$$\tau = \min\left\{ t \geq 0, \|S_t\|_{\widetilde{V}_t^{-1}} \geq \sqrt{2\log\left(\frac{1}{\delta}\right) + \log\left(\frac{\det(\widetilde{V}_t)}{\det(\mu_t I_d)}\right)} \right\}.$$

Thus,

$$\mathbb{P}\left( \exists t \geq 0, \|S_t\|_{\widetilde{V}_t^{-1}} \geq \sqrt{2\log\left(\frac{1}{\delta}\right) + \log\left(\frac{\det(\widetilde{V}_t)}{\det(\mu_t I_d)}\right)} \right) = \mathbb{P}(\tau < \infty)$$

$$= \mathbb{P}\left( \tau < \infty, \|S_\tau\|_{\widetilde{V}_\tau^{-1}} \geq \sqrt{2\log\left(\frac{1}{\delta}\right) + \log\left(\frac{\det(\widetilde{V}_\tau)}{\det(\mu_\tau I_d)}\right)} \right)$$

$$\leq \mathbb{P}\left( \|S_\tau\|_{\widetilde{V}_\tau^{-1}} \geq \sqrt{2\log\left(\frac{1}{\delta}\right) + \log\left(\frac{\det(\widetilde{V}_\tau)}{\det(\mu_\tau I_d)}\right)} \right)$$

$$= \mathbb{P}\left( \exp\left( \frac{1}{2}\|S_\tau\|^2_{\widetilde{V}_\tau^{-1}} \right) \sqrt{\frac{\det(\mu_\tau I_d)}{\det(\widetilde{V}_\tau)}} \geq \frac{1}{\delta} \right)$$

$$\leq \delta\mathbb{E}[\bar{M}_\tau(\mu_\tau)] \text{ (Markov's inequality)} \leq \delta \text{ (Lemma 3).}$$

$\square$

## A.2 Proof of Theorem 1

We recall that Theorem 1 is established in a stationary environment where $\forall t \geq 1, \theta_t^\star = \theta^\star$.

*Proof.* First note that,

$$\hat{\theta}_t = V_t^{-1} \sum_{s=1}^{t} w_s A_s X_s$$

$$= V_t^{-1} \sum_{s=1}^{t} w_s A_s (A_s^\top \theta^\star + \eta_s) \quad \text{(Equation 1)}$$

$$= V_t^{-1} \left( \sum_{s=1}^t w_s A_s A_s^\top \theta^\star + \lambda_t \theta^\star - \lambda_t \theta^\star \right) + V_t^{-1} S_t = \theta^\star - \lambda_t V_t^{-1} \theta^\star + V_t^{-1} S_t.$$

Thus,

$$\hat{\theta}_t - \theta^\star = V_t^{-1} S_t - \lambda_t V_t^{-1} \theta^\star. \qquad (7)$$

$\forall x \in \mathbb{R}^d, \forall t > 0$, we have

$$|x^\top (\hat{\theta}_t - \theta^\star)| \leq \|x\|_{V_t^{-1} \widetilde{V}_t V_t^{-1}} \left( \|V_t^{-1} S_t\|_{V_t \widetilde{V}_t^{-1} V_t} + \|\lambda_t V_t^{-1} \theta^\star\|_{V_t \widetilde{V}_t^{-1} V_t} \right)$$

$$\leq \|x\|_{V_t^{-1} \widetilde{V}_t V_t^{-1}} \left( \|S_t\|_{\widetilde{V}_t^{-1}} + \lambda_t \|\theta^\star\|_{\widetilde{V}_t^{-1}} \right).$$

By applying the previous inequality with $x = V_t \widetilde{V}_t^{-1} V_t (\hat{\theta}_t - \theta^\star)$, we have

$$\forall t, \|\hat{\theta}_t - \theta^\star\|_{V_t \widetilde{V}_t^{-1} V_t} \leq \|S_t\|_{\widetilde{V}_t^{-1}} + \lambda_t \|\theta^\star\|_{\widetilde{V}_t^{-1}}.$$

Knowing that $\widetilde{V}_t \geq \mu_t I_d$ and that $\widetilde{V}_t$ is positive definite, we have $\|\theta^\star\|_{\widetilde{V}_t^{-1}} \leq \frac{1}{\sqrt{\mu_t}} \|\theta^\star\|_2$.

Finally,

$$\forall t, \|\hat{\theta}_t - \theta^\star\|_{V_t \widetilde{V}_t^{-1} V_t} \leq \|S_t\|_{\widetilde{V}_t^{-1}} + \frac{\lambda_t}{\sqrt{\mu_t}} \|\theta^\star\|_2. \qquad (8)$$

From Proposition 1, we obtain the following any time high probability lower bound for $\|S_t\|_{\widetilde{V}_t^{-1}}$,

$$\mathbb{P} \left( \forall t \geq 0, \|S_t\|_{\widetilde{V}_t^{-1}} \leq \sigma \sqrt{2 \log \left( \frac{1}{\delta} \right) + \log \left( \frac{\det(\widetilde{V}_t)}{\mu_t^d} \right)} \right) \geq 1 - \delta.$$

Therefore by using inequality 8,

$$\mathbb{P} \left( \forall t \geq 0, \|\hat{\theta}_t - \theta^\star\|_{\widetilde{V}_t^{-1}} \leq \frac{\lambda_t}{\sqrt{\mu_t}} S + \sigma \sqrt{2 \log \left( \frac{1}{\delta} \right) + \log \left( \frac{\det(\widetilde{V}_t)}{\mu_t^d} \right)} \right) \geq 1 - \delta.$$

We obtain the exact formula of Theorem 1 by upper bounding $\det(\widetilde{V}_t)$ as proposed in Proposition 2 □

## B    D-LinUCB **Analysis**

In this section, the environment is non-stationary, which means that the unknown parameter $\theta^\star$ may evolve over time and is denoted $\theta_t^\star$. The reward generation process in the one presented in Equation (1).

### B.1    Preliminary results

In this section, $V_t$ and $\widetilde{V}_t$ are defined by

$$V_t = \sum_{s=1}^t \gamma^{-s} A_s A_s^\top + \lambda \gamma^{-t} I_d, \quad \widetilde{V}_t = \sum_{s=1}^t \gamma^{-2s} A_s A_s^\top + \lambda \gamma^{-2t} I_d.$$

We recall the definition of $\beta_t$:

$$\beta_t = \sqrt{\lambda} S + \sigma \sqrt{2 \log(1/\delta) + d \log \left( 1 + \frac{L^2(1 - \gamma^{2t})}{\lambda d (1 - \gamma^2)} \right)}.$$

With $\hat{\theta}_t$ defined in equation (3), the confidence ellipsoid we consider is defined by

$$\mathcal{C}_t = \left\{ \theta \in \mathbb{R}^d : \|\theta - \hat{\theta}_{t-1}\|_{V_{t-1} \widetilde{V}_{t-1}^{-1} V_{t-1}} \leq \beta_{t-1} \right\}. \qquad (9)$$

Theorem 1 can be applied with this choice of weights and regularization. We combine it with an upper bound for $\det(\widetilde{V}_t)$ given below.

**Proposition 2** (Determinant inequality for the weighted design matrix). *Let $(\lambda_t)_t$ be a deterministic sequence of regularization parameters. Let $V_t = \sum_{s=1}^{t} w_s A_s A_s^\top + \lambda_t I_d$ be the weighted design matrix. Under the assumption $\forall t, \|A_t\|_2 \leq L$, the following holds*

$$\det(V_t) \leq \left(\lambda_t + \frac{L^2 \sum_{s=1}^{t} w_s}{d}\right)^d.$$

*Proof.*

$$\det(V_t) = \prod_{i=1}^{d} l_i \quad (l_i \text{ are the eigenvalues})$$

$$\leq \left(\frac{1}{d}\sum_{i=1}^{d} l_i\right)^d \quad \text{(AM-GM inequality)}$$

$$\leq \left(\frac{1}{d}\text{trace}(V_t)\right)^d \leq \left(\frac{1}{d}\sum_{s=1}^{t} w_s \text{trace}(A_s A_s^\top) + \lambda_t\right)^d$$

$$\leq \left(\frac{1}{d}\sum_{s=1}^{t} w_s \|A_s\|_2^2 + \lambda_t\right)^d \leq \left(\lambda_t + \frac{L^2}{d}\sum_{s=1}^{t} w_s\right)^d.$$

$\square$

**Corollary 2.** *In the specific case where the weights are given by $w_t = \gamma^{-t}$ with $0 < \gamma < 1$. Proposition 2 can be rewritten*

$$\det(V_t) \leq \left(\lambda_t + \frac{L^2(\gamma^{-t} - 1)}{d(1 - \gamma)}\right)^d = \left(\lambda\gamma^{-t} + \frac{L^2(\gamma^{-t} - 1)}{d(1 - \gamma)}\right)^d.$$

We also have,

$$\det(\widetilde{V}_t) \leq \left(\mu_t + \frac{L^2(\gamma^{-2t} - 1)}{d(1 - \gamma^2)}\right)^d = \left(\lambda\gamma^{-2t} + \frac{L^2(\gamma^{-2t} - 1)}{d(1 - \gamma^2)}\right)^d.$$

*Proof.* Apply Proposition 2 and use $\sum_{s=1}^{t} \gamma^{-s} = \frac{\gamma^{-t}-1}{1-\gamma}$ and $\sum_{s=1}^{t} \gamma^{-2s} = \frac{\gamma^{-2t}-1}{1-\gamma^2}$. $\square$

Corollary 2 and Proposition 1 yield the following result.

**Corollary 3.** *$\forall \delta > 0$, with the weights $w_t = \gamma^{-t}$ and $0 < \gamma < 1$, we have*

$$\mathbb{P}\left(\exists t \geq 0, \|S_t\|_{\widetilde{V}_t^{-1}} \geq \sigma\sqrt{2\log\left(\frac{1}{\delta}\right) + d\log\left(1 + \frac{L^2(1 - \gamma^{2t})}{\lambda d(1 - \gamma^2)}\right)}\right) \leq \delta.$$

Thanks to this corollary we are now ready to show that $\bar{\theta}_t$ belongs to $\mathcal{C}_{t-1}$ with high probability.

**Proposition 3.** *Let $\mathcal{C}_t = \left\{\theta \in \mathbb{R}^d : \|\theta - \hat{\theta}_{t-1}\|_{V_{t-1}\widetilde{V}_{t-1}^{-1}V_{t-1}} \leq \beta_{t-1}\right\}$ denote the confidence ellipsoid. Let $\bar{\theta}_t = V_{t-1}^{-1}\left(\sum_{s=1}^{t-1} \gamma^{-s} A_s A_s^\top \theta_s^\star + \lambda\gamma^{-(t-1)}\theta_t^\star\right)$. Then, $\forall \delta > 0$,*

$$\mathbb{P}\left(\forall t \geq 1, \bar{\theta}_t \in \mathcal{C}_t\right) \geq 1 - \delta.$$

*Proof.*

$$\bar{\theta}_t - \hat{\theta}_{t-1} = V_{t-1}^{-1}\left(\sum_{s=1}^{t-1} \gamma^{-s} A_s A_s^\top \theta_s^\star + \lambda\gamma^{-(t-1)}\theta_t^\star - \sum_{s=1}^{t-1} \gamma^{-s} A_s X_s\right)$$

$$= V_{t-1}^{-1}\left(\sum_{s=1}^{t-1}\gamma^{-s}A_sA_s^\top\theta_s^\star + \lambda\gamma^{-(t-1)}\theta_t^\star - \sum_{s=1}^{t-1}\gamma^{-s}A_sA_s^\top\theta_s^\star - \sum_{s=1}^{t-1}\gamma^{-s}A_s\eta_s\right)$$

$$= -V_{t-1}^{-1}S_{t-1} + \lambda\gamma^{-(t-1)}V_{t-1}^{-1}\theta_t^\star.$$

Therefore,

$$\begin{aligned}
\|\bar{\theta}_t - \hat{\theta}_{t-1}\|_{V_{t-1}\widetilde{V}_{t-1}^{-1}V_{t-1}} &\leq \|S_{t-1}\|_{\widetilde{V}_{t-1}^{-1}} + \lambda\gamma^{-(t-1)}\|\theta_t^\star\|_{\widetilde{V}_{t-1}^{-1}} \\
&\leq \|S_{t-1}\|_{\widetilde{V}_{t-1}^{-1}} + \sqrt{\lambda}S \quad (\widetilde{V}_{t-1}^{-1} \leq 1/(\gamma^{-2(t-1)}\lambda)I_d \text{ and } \|\theta_t^\star\|_2 \leq S) \\
&\leq \beta_{t-1} \quad \text{(Corollary 3)}.
\end{aligned}$$

$\square$

## B.2 Control of the norm of actions

**Lemma 4.** *Let* $V_t = \sum_{s=1}^t \gamma^{-s}A_sA_s^\top + \lambda\gamma^{-t}I_d$ *and* $\widetilde{V}_t = \sum_{s=1}^t \gamma^{-2s}A_sA_s^\top + \lambda\gamma^{-2t}I_d$ *and* $0 < \gamma < 1$. *We have*

$$\forall t, \ V_t^{-1}\widetilde{V}_t V_t^{-1} \leq \gamma^{-t}V_t^{-1}.$$

*Proof.*

$$\widetilde{V}_t = \sum_{s=1}^t \gamma^{-2s}A_sA_s^\top + \lambda\gamma^{-2t}I_d \leq \gamma^{-t}\sum_{s=1}^t \gamma^{-s}A_sA_s^\top + \lambda\gamma^{-2t}I_d = \gamma^{-t}V_t.$$

Consequently,

$$V_t^{-1}\widetilde{V}_t V_t^{-1} \leq \gamma^{-t}V_t^{-1}V_t V_t^{-1} \leq \gamma^{-t}V_t^{-1}.$$

$\square$

Thanks to Lemma 4 we establish the following proposition,

**Proposition 4.**

$$\sum_{t=1}^T \min\left(1, \|A_t\|_{V_{t-1}^{-1}\widetilde{V}_{t-1}V_{t-1}^{-1}}^2\right) \leq 2\sum_{t=1}^T \log\left(1 + \gamma^{-t}\|A_t\|_{V_{t-1}^{-1}}^2\right) \leq 2\log\left(\frac{\det(V_T)}{\lambda^d}\right).$$

*Proof.* We first use the fact that: $\forall x \geq 0, \min(1, x) \leq 2\log(1 + x)$.

$$\begin{aligned}
\min\left(1, \|A_t\|_{V_{t-1}^{-1}\widetilde{V}_{t-1}V_{t-1}^{-1}}^2\right) &\leq 2\log\left(1 + \|A_t\|_{V_{t-1}^{-1}\widetilde{V}_{t-1}V_{t-1}^{-1}}^2\right) \\
&\leq 2\log\left(1 + \gamma^{-(t-1)}\|A_t\|_{V_{t-1}^{-1}}^2\right) \quad \text{(Lemma 4)} \\
&\leq 2\log\left(1 + \gamma^{-t}\|A_t\|_{V_{t-1}^{-1}}^2\right) \quad (\gamma \leq 1).
\end{aligned}$$

Furthermore,

$$V_t \geq \gamma^{-t}A_tA_t^\top + V_{t-1} \geq V_{t-1}^{1/2}(I_d + \gamma^{-t}V_{t-1}^{-1/2}A_tA_t^\top V_{t-1}^{-1/2})V_{t-1}^{1/2}.$$

Given that all those matrices are symmetric positive definite, the previous inequality implies that

$$\begin{aligned}
\det(V_t) &\geq \det(V_{t-1})\det(1 + (\gamma^{-t/2}V_{t-1}^{-1/2}A_t)(\gamma^{-t/2}V_{t-1}^{-1/2}A_t)^\top) \\
&\geq \det(V_{t-1})\left(1 + \gamma^{-t}\|A_t\|_{V_{t-1}^{-1}}^2\right) \quad \left(\text{Using } \det(I_d + xx^\top) = 1 + \|x\|_2^2\right).
\end{aligned}$$

Therefore,

$$\frac{\det(V_T)}{\det(V_0)} = \prod_{t=1}^T \frac{\det(V_t)}{\det(V_{t-1})} \geq \prod_{t=1}^T(1 + \gamma^{-t}\|A_t\|_{V_{t-1}^{-1}}^2).$$

Finally by applying the log function to the previous inequality,

$$\sum_{t=1}^{T} \min\left(1, \|A_t\|^2_{V_{t-1}^{-1}\widetilde{V}_{t-1}V_{t-1}^{-1}}\right) \leq 2\sum_{t=1}^{T} \log\left(1 + \gamma^{-t}\|A_t\|^2_{V_{t-1}^{-1}}\right) \leq 2\log\left(\frac{\det(V_T)}{\det(V_0)}\right).$$

$\square$

**Corollary 4.**

$$\sqrt{\sum_{t=1}^{T} \min\left(1, \|A_t\|^2_{V_{t-1}^{-1}\widetilde{V}_{t-1}V_{t-1}^{-1}}\right)} \leq \sqrt{2d}\sqrt{T\log\left(\frac{1}{\gamma}\right) + \log\left(1 + \frac{L^2}{d\lambda(1-\gamma)}\right)}.$$

*Proof.* The proof of this corollary is based on the previous lemma and on Corollary 2. We have

$$\log\left(\frac{\det(V_T)}{\det(V_0)}\right) \leq \log\left(\frac{1}{\lambda^d}\left(\lambda\gamma^{-T} + \frac{L^2(\gamma^{-T}-1)}{d(1-\gamma)}\right)^d\right) \quad \text{(Corollary 2)}$$

$$\leq dT\log\left(\frac{1}{\gamma}\right) + d\log\left(1 + \frac{L^2}{d\lambda(1-\gamma)}\right).$$

$\square$

## B.3 Proof of Theorem 2

In this subsection we give the proof of Theorem 2 for the high probability upper-bound of the regret for D-LinUCB.

*Proof.*

First step: Upper bound for the instantaneous regret.

Let $A_t^{\star} = \arg\max_{a \in \mathcal{A}_t}\langle a, \theta_t^{\star}\rangle$ and $\theta_t = \arg\max_{\theta \in \mathcal{C}_t}\langle A_t, \theta\rangle$. We have,

$$r_t = \max_{a \in \mathcal{A}_t}\langle a, \theta_t^{\star}\rangle - \langle A_t, \theta_t^{\star}\rangle = \langle A_t^{\star} - A_t, \theta_t^{\star}\rangle$$

$$= \langle A_t^{\star} - A_t, \bar{\theta}_t\rangle + \langle A_t^{\star} - A_t, \theta_t^{\star} - \bar{\theta}_t\rangle.$$

Under the event $\{\forall t > 0, \bar{\theta}_t \in \mathcal{C}_t\}$, that occurs with probability at least $1 - \delta$ thanks to Proposition 3, we have,

$$\langle A_t^{\star}, \bar{\theta}_t\rangle \leq \arg\max_{\theta \in \mathcal{C}_t}\langle A_t^{\star}, \theta\rangle = \text{UCB}_t(A_t^{\star}) \leq \text{UCB}_t(A_t) = \arg\max_{\theta \in \mathcal{C}_t}\langle A_t, \theta\rangle = \langle A_t, \theta_t\rangle. \quad (10)$$

Then, with probability at least $1 - \delta$, $\forall t > 0$,

$$r_t \leq \langle A_t, \theta_t - \bar{\theta}_t\rangle + \langle A_t^{\star} - A_t, \theta_t^{\star} - \bar{\theta}_t\rangle$$

$$\leq \|A_t\|_{V_{t-1}^{-1}\widetilde{V}_{t-1}V_{t-1}^{-1}}\|\theta_t - \bar{\theta}_t\|_{V_{t-1}\widetilde{V}_{t-1}^{-1}V_{t-1}} + \|A_t^{\star} - A_t\|_2\|\theta_t^{\star} - \bar{\theta}_t\|_2 \quad \text{(Cauchy-Schwarz)}$$

$$\leq \|A_t\|_{V_{t-1}^{-1}\widetilde{V}_{t-1}V_{t-1}^{-1}}\|\theta_t - \bar{\theta}_t\|_{V_{t-1}\widetilde{V}_{t-1}^{-1}V_{t-1}} + 2L\|\theta_t^{\star} - \bar{\theta}_t\|_2 \quad (\forall a \in \mathcal{A}_t \|a\|_2 \leq L).$$

As discussed in Section 3.2, the two terms are upper bounded using different techniques. The first term is handled with the equivalent in a non-stationary environment of the deviation inequality of Theorem 1 and the second term is the equivalent of the bias.

Second step: Upper bound for $\|\theta_t - \bar{\theta}_t\|_{V_{t-1}\widetilde{V}_{t-1}^{-1}V_{t-1}}$.

We have,

$$\|\theta_t - \bar{\theta}_t\|_{V_{t-1}\widetilde{V}_{t-1}^{-1}V_{t-1}} \leq \|\theta_t - \hat{\theta}_{t-1}\|_{V_{t-1}\widetilde{V}_{t-1}^{-1}V_{t-1}} + \|\bar{\theta}_t - \hat{\theta}_{t-1}\|_{V_{t-1}\widetilde{V}_{t-1}^{-1}V_{t-1}} \leq 2\beta_{t-1},$$

where the last inequality holds because under our assumption $\bar{\theta}_t \in \mathcal{C}_t$ with high probability and by definition $\theta_t \in \mathcal{C}_t$.

Third step: Upper bound for the bias.

Let $D > 0$,

$$\|\theta_t^\star - \bar{\theta}_t\|_2 = \|V_{t-1}^{-1} \sum_{s=1}^{t-1} \gamma^{-s} A_s A_s^\top (\theta_s^\star - \theta_t^\star)\|_2$$

$$\leq \| \sum_{s=t-D}^{t-1} V_{t-1}^{-1} \gamma^{-s} A_s A_s^\top (\theta_s^\star - \theta_t^\star)\|_2 + \|V_{t-1}^{-1} \sum_{s=1}^{t-D-1} \gamma^{-s} A_s A_s^\top (\theta_s^\star - \theta_t^\star)\|_2$$

$$\leq \| \sum_{s=t-D}^{t-1} V_{t-1}^{-1} \gamma^{-s} A_s A_s^\top \sum_{p=s}^{t-1} (\theta_p^\star - \theta_{p+1}^\star)\|_2 + \| \sum_{s=1}^{t-D-1} \gamma^{-s} A_s A_s^\top (\theta_s^\star - \theta_t^\star)\|_{V_{t-1}^{-2}}$$

$$\leq \| \sum_{p=t-D}^{t-1} V_{t-1}^{-1} \gamma^{-s} A_s A_s^\top \sum_{s=t-D}^{p} (\theta_p^\star - \theta_{p+1}^\star)\|_2 + \frac{1}{\lambda} \sum_{s=1}^{t-D-1} \gamma^{t-1-s} \|A_s A_s^\top (\theta_s^\star - \theta_t^\star)\|_2$$

$$\leq \sum_{p=t-D}^{t-1} \|V_{t-1}^{-1} \sum_{s=t-D}^{p} \gamma^{-s} A_s A_s^\top (\theta_p^\star - \theta_{p+1}^\star)\|_2 + \frac{2L^2 S}{\lambda} \sum_{s=1}^{t-D-1} \gamma^{t-1-s}$$

$$\leq \sum_{p=t-D}^{t-1} \lambda_{\max} \left( V_{t-1}^{-1} \sum_{s=t-D}^{p} \gamma^{-s} A_s A_s^\top \right) \|\theta_p^\star - \theta_{p+1}^\star\|_2 + \frac{2L^2 S}{\lambda} \frac{\gamma^D}{1-\gamma}.$$

The first inequality is a consequence of the triangular inequality. The third inequality uses that $V_{t-1}^{-2} \leq (\frac{\gamma^{t-1}}{\lambda})^2 I_d$. In the last inequality, we have used the fact that for a symmetric matrix $M \in \mathcal{M}_d(\mathbb{R})$ and a vector $x \in \mathbb{R}^d$, $\|Mx\|_2 \leq \lambda_{\max}(M)\|x\|_2$.

Furthermore, for $x$ such that $\|x\|_2 \leq 1$, we have that for $t - D \leq p \leq t - 1$,

$$x^\top V_{t-1}^{-1} \sum_{s=t-D}^{p} \gamma^{-s} A_s A_s^\top x \leq x^\top V_{t-1}^{-1} \sum_{s=1}^{t-1} \gamma^{-s} A_s A_s^\top x + \lambda \gamma^{-(t-1)} x^\top V_{t-1}^{-1} x$$

$$\leq x^\top V_{t-1}^{-1} (\sum_{s=1}^{t-1} \gamma^{-s} A_s A_s^\top + \lambda \gamma^{-(t-1)} I_d) x = x^\top x \leq 1.$$

Therefore, for all $p$ such that $t - D \leq p \leq t - 1$, $\lambda_{\max} \left( V_{t-1}^{-1} \sum_{s=t-D}^{p} \gamma^{-s} A_s A_s^\top \right) \leq 1$.

By combining the second and the third step, with probability at least $1 - \delta$:

$$r_t \leq 2L \sum_{p=t-D}^{t-1} \|\theta_p^\star - \theta_{p+1}^\star\|_2 + \frac{4L^3 S}{\lambda} \frac{\gamma^D}{1-\gamma} + 2\beta_{t-1} \|A_t\|_{V_{t-1}^{-1} \tilde{V}_{t-1} V_{t-1}^{-1}}.$$

The assumption $|\langle A_t, \theta_t^\star \rangle| \leq 1$ also implies $r_t \leq 2$. Hence, with probability at least $1 - \delta$:

$$r_t \leq 2L \sum_{p=t-D}^{t-1} \|\theta_p^\star - \theta_{p+1}^\star\|_2 + 4L^3 S \frac{\gamma^D}{1-\gamma} + 2\beta_{t-1} \min(1, \|A_t\|_{V_{t-1}^{-1} \tilde{V}_{t-1} V_{t-1}^{-1}}). \qquad (11)$$

To conclude the proof we use the results of Subsection B.2.

Final step:

$$R_T = \sum_{t=1}^{T} r_t$$

$$\leq 2L \sum_{t=1}^{T} \sum_{p=t-D}^{t-1} \|\theta_p^\star - \theta_{p+1}^\star\|_2 + \frac{4L^3 S}{\lambda} \frac{\gamma^D}{1-\gamma} T + 2\beta_T \sum_{t=1}^{T} \min \left( 1, \|A_t\|_{V_{t-1}^{-1} \tilde{V}_{t-1} V_{t-1}^{-1}} \right)$$

$$\leq 2L \sum_{t=1}^{T} \sum_{p=t-D}^{t-1} \|\theta_p^\star - \theta_{p+1}^\star\|_2 + \frac{4L^3 S}{\lambda} \frac{\gamma^D}{1-\gamma} T + 2\beta_T \sqrt{T} \sqrt{\sum_{t=1}^{T} \min \left( 1, \|A_t\|^2_{V_{t-1}^{-1} \tilde{V}_{t-1} V_{t-1}^{-1}} \right)}$$

$$\leq 2LB_T D + \frac{4L^3 S}{\lambda} \frac{\gamma^D}{1-\gamma} T + 2\sqrt{2}\beta_T \sqrt{dT} \sqrt{T \log(1/\gamma) + \log\left(1 + \frac{L^2}{d\lambda(1-\gamma)}\right)}.$$

In the first inequality, we use that $t \mapsto \beta_t$ is increasing. The second inequality is an application of the Cauchy-Schwarz inequality to the third term and the last inequality is an application of Corollary 4. □

### B.4 Proof of Corollary 1

*Proof.* Let $\gamma$ be defined as $\gamma = 1 - (\frac{B_T}{dT})^{2/3}$ and $D = \frac{\log(T)}{(1-\gamma)}$. With this choice of $\gamma$, $D$ is equivalent to $d^{2/3} B_T^{-2/3} T^{2/3} \log(T)$. Thus, $DB_T$ is equivalent to $d^{2/3} B_T^{1/3} T^{2/3} \log(T)$.

In addition,

$$\gamma^D = \exp(D \log(\gamma)) = \exp\left(\frac{\log(\gamma)}{1-\gamma} \log(T)\right) \sim 1/T.$$

Hence, $T\gamma^D \frac{1}{1-\gamma}$ behaves as $d^{2/3} T^{2/3} B_T^{-2/3}$.

Furthermore, $\log(1/\gamma) \sim d^{-2/3} B_T^{2/3} T^{-2/3}$, implying that $T \log(1/\gamma) \sim d^{-2/3} B_T^{2/3} T^{1/3}$.

As a result, it holds that, $\beta_T \sqrt{dT} \sqrt{T \log(1/\gamma) + \log\left(1 + \frac{L^2}{d\lambda(1-\gamma)}\right)}$ is equivalent to $dT^{1/2} \sqrt{\log(T/B_T)} \sqrt{d^{-2/3} B_T^{2/3} T^{1/3}} = d^{2/3} B_T^{1/3} T^{2/3} \sqrt{\log(T/B_T)}$.

By adding those three terms and neglecting the log factors, we obtain the desired result. □

## C  A new analysis of the `SW-LinUCB` algorithm

In this section we propose a new analysis of the `SW-LinUCB` algorithm. This is useful as the proof provided in [11] has several gaps. First, Lemma 2 of [11] is presented as a specific case of the analysis of [1]. It would hold in the case of a growing window, where the argument developed in [1] could be used, but not with a sliding window, where past actions are removed from the design matrix. Furthermore, Theorem 2 of [11] that bounds $|\langle x, \hat{\theta}_{t-1} - \theta_t^\star \rangle|$ for any fixed direction $x$ with high probability is used in equation (42) with $x$ replaced by $A_t$, whereas $A_t$ is a random variable strongly related to $\hat{\theta}_{t-1}$.

We only mention this analysis in the Appendix because the deviation inequalities established for the weighted model can not be used. Nevertheless, we believe that this analysis gives new insights on the problem with a sliding window.

### C.1  Deviation inequality

Let us introduce some notations to clarify the model. We suppose that there is a sliding window of length $l$, such that the estimate of the unknown parameter at time $t$ is based on the $l$ last observations. The optimization program solved is

$$\hat{\theta}_t = \arg\min_{\theta \in \mathbb{R}^d} \left( \sum_{s=\max(1,t-l+1)}^{t} (X_s - \langle A_s, \theta \rangle)^2 + \lambda \|\theta\|_2^2 \right).$$

One has

$$\hat{\theta}_t = V_t^{-1} \sum_{s=\max(1,t-l+1)}^{t} A_s X_s, \quad \text{where} \quad V_t = \sum_{s=\max(1,t-l+1)}^{t} A_s A_s^\top + \lambda I_d. \tag{12}$$

The expression linking the matrices $V_t$ and $V_{t-1}$ is the following

$$V_t = V_{t-1} + A_t A_t^\top - A_{t-l} A_{t-l}^\top.$$

The specificity of the sliding window model is that at time $t$, to update the design matrix, a new action vector $A_t$ is added but the oldest term $A_{t-l}$ is also removed . When considering the equivalent of the quantity $M_t(x)$ defined in the Appendix A, the property of supermartingale does not hold anymore because of this loss of information. For this reason, all the reasoning that was done in [1] can not be applied directly.

The reward generation process we consider is still the one presented in Equation 1. As for the D-LinUCB model, the results are stated with $\sigma$-subgaussian random noises but the proofs are done with $\sigma = 1$. Let $S_t = \sum_{s=\max(1,t-l+1)}^{t} A_s \eta_s$. We start by giving the proof of the analogue of Lemma 2 presented in [11]. We give an instantaneous deviation inequality.

**Proposition 5** (Instantaneous deviation inequality with a sliding window). *Let $t$ be a fixed time instant. For all $\delta > 0$,*

$$\mathbb{P}\left( \|S_t\|_{V_t^{-1}} \geq \sigma\sqrt{2\log\left(\frac{1}{\delta}\right) + \log\left(\frac{\det(V_t)}{\lambda^d}\right)} \right) \leq \delta.$$

*Proof.* We present an interesting trick in this proof for avoiding the loss of information due to the sliding window that is only usable for instantaneous deviation inequalities.

Let $t$ be the time instant of interest. We assume that $t \geq l$. We know that the estimate $\hat{\theta}_t$ is only based on observations between time $t - l + 1$ to $t$. The trick is to create a fictive regression model starting a time $t - l$ and receiving the exact same information as the true model between the time instants $t - l + 1$ to $t$.

To ease the understanding of the proof, the notations with dotted symbols refer to the fictive model. Let $u$ be a time instant in $[\![t-l, t]\!]$. Let $\dot{V}_u = \sum_{\max(1,t-l+1)}^{u} A_s A_s^\top + \lambda I_d$, $\dot{S}_u = \sum_{\max(1,t-l+1)}^{u} A_s \eta_s$ and $\dot{M}_u(x) = \exp(x^\top \dot{S}_u - x^\top \dot{V}_u(0) x / 2)$. Once again, $\dot{V}_u(0) = \sum_{\max(1,t-l+1)}^{u} A_s A_s^\top$ corresponds to the design matrix without the regularization term. By definition, $\forall x \in \mathbb{R}^d, \dot{M}_{t-l}(x) = 1$.

Using the 1-subgaussianity and following the lines of the proof of Lemma 1,

$$\mathbb{E}[\dot{M}_u(x)|\mathcal{F}_{u-1}] \leq \dot{M}_{u-1}(x).$$

Therefore, $\forall u \in [\![t - l, t]\!], \mathbb{E}[\dot{M}_u(x)] \leq \mathbb{E}[\dot{M}_{t-l}(x)] = 1$. In particular for $u = t$, $\forall x \in \mathbb{R}^d, \mathbb{E}[\dot{M}_t(x)] \leq 1$. By introducing a measure of probability $h = \mathcal{N}(0, \frac{1}{\lambda}I_d)$, we still have $\mathbb{E}\left[\int \dot{M}_t(x)dh(x)\right] \leq 1$ using a similar reasoning than in Lemma 2. We can also give an exact formula for $\int \dot{M}_t(x)dh(x)$ with the chosen $h$. Let us remark that $\dot{S}_t = S_t$ and $\dot{V}_t = V_t$.

$$\int_{\mathbb{R}^d} \dot{M}_t(x)dh(x) = \frac{1}{\sqrt{(2\pi)^d \det(1/\lambda I_d)}} \int_{\mathbb{R}^d} \exp\left( x^\top S_t - \frac{1}{2}\|x\|_{\lambda I_d}^2 - \frac{1}{2}\|x\|_{V_t(0)}^2 \right) dx$$

$$= \frac{1}{\sqrt{(2\pi)^d \det(1/\lambda I_d)}} \int_{\mathbb{R}^d} \exp\left( 1/2\|S_t\|_{V_t^{-1}}^2 - 1/2\|x - V_t^{-1}S_t\|_{V_t}^2 \right) dx$$

$$= \frac{\exp\left(\frac{1}{2}\|S_t\|_{V_t^{-1}}^2\right)}{\sqrt{(2\pi)^d \det(1/\lambda I_d)}} \int_{\mathbb{R}^d} \exp\left( -\frac{1}{2}\|x - V_t^{-1}S_t\|_{V_t}^2 \right) dx$$

$$= \frac{\exp\left(\frac{1}{2}\|S_t\|_{V_t^{-1}}^2\right)}{\sqrt{(2\pi)^d \det(1/\lambda I_d)}} \sqrt{(2\pi)^d \det\left(V_t^{-1}\right)}$$

$$= \exp\left(\frac{1}{2}\|S_t\|_{V_t^{-1}}^2\right) \sqrt{\frac{\det(\lambda I_d)}{\det(V_t)}}.$$

For this reason,

$$\mathbb{P}\left( \|S_t\|_{V_t^{-1}} \geq \sqrt{2\log\left(\frac{1}{\delta}\right) + \log\left(\frac{\det(V_t)}{\det(\lambda I_d)}\right)} \right)$$

$$= \mathbb{P}\left(\exp\left(\frac{1}{2}\|S_t\|_{V_t^{-1}}^2\right)\sqrt{\frac{\det(\lambda I_d)}{\det(V_t)}} \geq \frac{1}{\delta}\right)$$

$$\leq \delta\mathbb{E}\left[\int_{\mathbb{R}^d} \dot{M}_t(x)dh(x)\right] \quad \text{(Markov's inequality)}$$

$$\leq \delta.$$

$\square$

The next step is to upper-bound the quantity $\det(V_t)$ similarly as in Proposition 2 for the weighted model.

**Proposition 6** (Determinant inequality for the design matrix with a sliding window)**.** *In the specific case where $V_t$ is defined as $V_t = \sum_{s=\max(1,t-l+1)}^{t} A_s A_s^\top + \lambda I_d$. Under the assumption $\forall t, \|A_t\|_2 \leq L$, the following holds,*

$$\det(V_t) \leq \left(\lambda + \frac{L^2 \min(t,l)}{d}\right)^d.$$

The proof of this proposition is the same as in Proposition 2. By using the previous inequality, we can obtain the following proposition,

**Proposition 7.** *When using a sliding window model where the last $l$ terms are considered, for all $\delta > 0$,*

$$\mathbb{P}\left(\exists t \leq T, \|S_t\|_{V_t^{-1}} \geq \sigma\sqrt{2\log\left(\frac{T}{\delta}\right) + d\log\left(1 + \frac{L^2 \min(t,l)}{\lambda d}\right)}\right) \leq \delta.$$

*Proof.*

$$\mathbb{P}\left(\exists t \leq T, \|S_t\|_{V_t^{-1}} \geq \sigma\sqrt{2\log\left(\frac{T}{\delta}\right) + d\log\left(1 + \frac{L^2 \min(t,l)}{\lambda d}\right)}\right)$$

$$\leq \sum_{t=1}^{T} \mathbb{P}\left(\|S_t\|_{V_t^{-1}} \geq \sigma\sqrt{2\log\left(\frac{T}{\delta}\right) + d\log\left(1 + \frac{L^2 \min(t,l)}{\lambda d}\right)}\right)$$

$$\leq \sum_{t=1}^{T} \mathbb{P}\left(\|S_t\|_{V_t^{-1}} \geq \sigma\sqrt{2\log\left(\frac{T}{\delta}\right) + \log\left(\frac{\det(V_t)}{\lambda^d}\right)}\right)$$

$$\leq \sum_{t=1}^{T} \frac{\delta}{T} \quad \text{(Proposition 5)} \leq \delta.$$

$\square$

## C.2 Regret analysis

The regret analysis of the `SW-LinUCB` algorithm is similar to the one proposed for `D-LinUCB`. We start by defining the confidence ellipsoid used by the algorithm `SW-LinUCB`.

With the `SW-LinUCB` algorithm, the $\beta_t$ term is defined in the following way,

$$\beta_t = \sqrt{\lambda}S + \sigma\sqrt{2\log\left(\frac{T}{\delta}\right) + d\log\left(1 + \frac{L^2 \min(t,l)}{\lambda d}\right)} \tag{13}$$

Remark: The cost of loosing some information at each step due to the sliding window when $t > l$ is the term $\log\left(\frac{T}{\delta}\right)$ rather than $\log\left(\frac{1}{\delta}\right)$ in the definition of $\beta_t$.

Note that due to the use of a union bound technique the confidence radius is larger than the one suggested in [11]. Nevertheless, this was not taken into account in simulations for `SW-LinUCB`.

**Proposition 8.** *Let* $\mathcal{C}_t = \left\{ \theta \in \mathbb{R}^d : \|\theta - \hat{\theta}_{t-1}\|_{V_{t-1}^{-1}} \leq \beta_{t-1} \right\}$ *denote the confidence ellipsoid. Let*
$\bar{\theta}_t = V_{t-1}^{-1} \left( \sum_{s=\max(1,t-l)}^{t-1} A_s A_s^\top \theta_s^\star + \lambda \theta_t^\star \right)$. *Then*, $\forall \delta > 0$,

$$\mathbb{P}\left( \forall t \geq 1, \bar{\theta}_t \in \mathcal{C}_t \right) \geq 1 - \delta.$$

*Proof.*

$$\bar{\theta}_t - \hat{\theta}_{t-1} = V_{t-1}^{-1} \left( \sum_{s=\max(1,t-l)}^{t-1} A_s A_s^\top \theta_s^\star + \lambda \theta_t^\star - \sum_{s=\max(1,t-l)}^{t-1} A_s A_s^\top \theta_s^\star - \sum_{s=\max(1,t-l)}^{t-1} A_s \eta_s \right)$$

$$= -V_{t-1}^{-1} S_{t-1} + \lambda V_{t-1}^{-1} \theta_t^\star.$$

Therefore,

$$\|\bar{\theta}_t - \hat{\theta}_{t-1}\|_{V_{t-1}^{-1}} \leq \|S_{t-1}\|_{V_{t-1}^{-1}} + \lambda \|\theta_t^\star\|_{V_{t-1}^{-1}}$$

$$\leq \|S_{t-1}\|_{V_{t-1}^{-1}} + \sqrt{\lambda} S \quad (V_{t-1}^{-1} \leq \frac{1}{\lambda} I_d)$$

$$\leq \beta_{t-1} \quad \text{(with probability} \geq 1 - \delta \text{ thanks to Proposition 7).}$$

$\square$

We need to bound the quantity $\sum_{t=1}^T \min\left(1, \|A_t\|_{V_{t-1}^{-1}}^2\right)$. An analysis of this quantity is already proved in [11]. Nevertheless, we provide a simpler analysis in the following proposition.

**Proposition 9.** *With the sliding window model, the following upper bound holds,*

$$\sum_{t=1}^T \min\left(1, \|A_t\|_{V_{t-1}^{-1}}^2\right) \leq 2d\lceil T/l \rceil \log\left(1 + \frac{lL^2}{\lambda d}\right).$$

*Proof.* We start by rewriting the sum as follows.

$$\sum_{t=1}^T \min\left(1, \|A_t\|_{V_{t-1}^{-1}}^2\right) = \sum_{k=0}^{\lceil T/l \rceil - 1} \sum_{t=kl+1}^{(k+1)l} \min\left(1, \|A_t\|_{V_{t-1}^{-1}}^2\right)$$

For the $k$-th block of length $l$ we define the matrix $W_t^{(k)} = \sum_{s=kl+1}^t A_s A_s^\top + \lambda I_d$. We also have $\forall t \in [\![ kl, (k+1)l ]\!], V_t \geq W_t^{(k)}$ as every term in $W_t^{(k)}$ is contained in $V_t$ and the extra-terms in $V_t$ correspond to positive definite matrices. The matrices are definite positive, thus $V_t^{-1} \leq (W_t^{(k)})^{-1}$ and consequently,

$$\sum_{k=0}^{\lceil T/l \rceil - 1} \sum_{t=kl+1}^{(k+1)l} \min\left(1, \|A_t\|_{V_{t-1}^{-1}}^2\right) \leq \sum_{k=0}^{\lceil T/l \rceil - 1} \sum_{t=kl+1}^{(k+1)l} \min\left(1, \|A_t\|_{(W_{t-1}^{(k)})^{-1}}^2\right)$$

Furthermore, $\forall t \in [\![ kl, (k+1)l ]\!]$ we have,

$$\det(W_t^{(k)}) = \det(W_{t-1}^{(k)}) \left(1 + \|A_t\|_{(W_{t-1}^{(k)})^{-1}}^2\right).$$

With positive definitive matrices whose determinants are strictly positive, this implies that

$$\frac{\det(W_{(k+1)l}^{(k)})}{\det(W_{kl}^{(k)})} = \prod_{t=kl+1}^{(k+1)l} \frac{\det(W_t^{(k)})}{\det(W_{t-1}^{(k)})} = \prod_{t=kl+1}^{(k+1)l} \left(1 + \|A_t\|_{(W_{t-1}^{(k)})^{-1}}^2\right).$$

By definition we have $W_{kl}^{(k)} = \lambda I_d$ and $\forall x \geq 0, \min(1,x) \leq 2\log(1+x)$. So,

$$\sum_{t=1}^T \min\left(1, \|A_t\|_{V_{t-1}^{-1}}^2\right) \leq 2 \sum_{k=0}^{\lceil T/l \rceil - 1} \sum_{t=kl+1}^{(k+1)l} \log\left(1 + \|A_t\|_{(W_{t-1}^{(k)})^{-1}}^2\right)$$

$$\leq 2 \sum_{k=0}^{\lceil T/l \rceil - 1} \log \left( \frac{\det(W_{(k+1)l}^{(k)})}{\lambda^d} \right).$$

Knowing that $W_{(k+1)l}^{(k)}$ contains exactly $l$ terms allows us to give the following bound (by following the proof of Proposition 2),

$$\det(W_{(k+1)l}^{(k)}) \leq \left( \lambda + \frac{L^2 l}{d} \right)^d.$$

Finally,

$$\sum_{t=1}^{T} \min \left( 1, \|A_t\|_{V_{t-1}^{-1}}^2 \right) \leq 2d \lceil T/l \rceil \log \left( 1 + \frac{L^2 l}{\lambda d} \right).$$

$\square$

With those results we can give a high probability upper bound for the cumulative dynamic regret of the SW-LinUCB algorithm.

**Theorem 3.** *Assuming that $\sum_{s=1}^{T-1} \|\theta_s^\star - \theta_{s+1}^\star\|_2 \leq B_T$, the regret of the SW-LinUCB algorithm may be bounded for all $l > 0$, with probability at least $1 - \delta$, by*

$$R_T \leq 2LB_T l + 2\sqrt{2}\beta_T \sqrt{dT} \sqrt{\lceil T/l \rceil} \sqrt{\log \left( 1 + \frac{L^2 l}{\lambda d} \right)},$$

*where $\beta_T$ is defined in Equation (13).*

*Proof.*

1rst step: Upper bound for the instantaneous regret

Defining $A_t^\star = \arg\max_{a \in \mathcal{A}_t} \langle a, \theta_t^\star \rangle$ and $\theta_t = \arg\max_{\theta \in \mathcal{C}_t} \langle A_t, \theta \rangle$. We have,

$$r_t = \max_{a \in \mathcal{A}_t} \langle a, \theta_t^\star \rangle - \langle A_t, \theta_t^\star \rangle = \langle A_t^\star - A_t, \theta_t^\star \rangle$$
$$= \langle A_t^\star - A_t, \bar{\theta}_t \rangle + \langle A_t^\star - A_t, \theta_t^\star - \bar{\theta}_t \rangle$$

Under the event $\{\forall t > 0, \bar{\theta}_t \in \mathcal{C}_t\}$, that occurs with probability at least $1 - \delta$ thanks to Proposition 8,

$$\langle A_t^\star, \bar{\theta}_t \rangle \leq \arg\max_{\theta \in \mathcal{C}_t} \langle A_t^\star, \theta \rangle = \mathrm{UCB}_t(A_t^\star) \leq \mathrm{UCB}_t(A_t) = \arg\max_{\theta \in \mathcal{C}_t} \langle A_t, \theta \rangle = \langle A_t, \theta_t \rangle \quad (14)$$

Using Inequality (14), with probability larger than $1 - \delta$, $\forall t > 0$,

$$r_t \leq \langle A_t, \theta_t - \bar{\theta}_t \rangle + \langle A_t^\star - A_t, \theta_t^\star - \bar{\theta}_t \rangle$$
$$\leq \|A_t\|_{V_{t-1}^{-1}} \|\theta_t - \bar{\theta}_t\|_{V_{t-1}} + \|A_t^\star - A_t\|_2 \|\theta_t^\star - \bar{\theta}_t\|_2 \quad \text{(Cauchy-Schwarz)}$$
$$\leq \|A_t\|_{V_{t-1}^{-1}} \|\theta_t - \bar{\theta}_t\|_{V_{t-1}} + 2L\|\theta_t^\star - \bar{\theta}_t\|_2 \quad \text{(Bounded action assumption)}.$$

As for the analysis of the regret for the D-LinUCB algorithm, the two terms are upper bounded using different techniques. The first term is handled with the deviation inequality of Proposition 8.

2nd step: Upper bound for $\|\theta_t - \bar{\theta}_t\|_{V_{t-1}}$

We have,

$$\|\theta_t - \bar{\theta}_t\|_{V_{t-1}} \leq \|\theta_t - \hat{\theta}_{t-1}\|_{V_{t-1}} + \|\bar{\theta}_t - \hat{\theta}_{t-1}\|_{V_{t-1}} \leq 2\beta_{t-1}.$$

Where the last inequality holds because under our assumption $\bar{\theta}_t \in \mathcal{C}_t$ with probability at least $1 - \delta$ and by definition $\theta_t \in \mathcal{C}_t$.

3rd step: Upper bound for the bias.

This step is similar to the proof proposed in [11] for Lemma 1.

$$\|\theta_t^\star - \bar{\theta}_t\|_2 = \left\| V_{t-1}^{-1} \left( \sum_{s=\max(1,t-l)}^{t-1} A_s A_s^\top (\theta_s^\star - \theta_t^\star) \right) \right\|_2$$

$$\leq \left\| \sum_{s=\max(1,t-l)}^{t-1} V_{t-1}^{-1} A_s A_s^\top \sum_{p=s}^{t-1} (\theta_p^\star - \theta_{p+1}^\star) \right\|_2$$

$$\leq \left\| \sum_{p=\max(1,t-l)}^{t-1} V_{t-1}^{-1} \sum_{s=\max(1,t-l)}^{p} A_s A_s^\top (\theta_p^\star - \theta_{p+1}^\star) \right\|_2$$

$$\leq \sum_{p=\max(1,t-l)}^{t-1} \left\| V_{t-1}^{-1} \sum_{s=\max(1,t-l)}^{p} A_s A_s^\top (\theta_p^\star - \theta_{p+1}^\star) \right\|_2$$

$$\leq \sum_{p=\max(1,t-l)}^{t-1} \lambda_{\max} \left( V_{t-1}^{-1} \sum_{s=\max(1,t-l)}^{p} A_s A_s^\top \right) \|\theta_p^\star - \theta_{p+1}^\star\|_2.$$

Furthermore, for $x \in \mathbb{R}^d$ such that $\|x\|_2 \leq 1$, we have that for $\max(1, t - l) \leq p \leq t - 1$,

$$x^\top V_{t-1}^{-1} \sum_{s=\max(1,t-l)}^{p} A_s A_s^\top x \leq x^\top V_{t-1}^{-1} \sum_{s=\max(1,t-l)}^{t-1} A_s A_s^\top x + \lambda x^\top V_{t-1}^{-1} x$$

$$\leq x^\top V_{t-1}^{-1} \left( \sum_{s=\max(1,t-l)}^{t-1} A_s A_s^\top + \lambda I_d \right) x = x^\top x \leq 1.$$

By combining the second and the third step,

$$r_t \leq 2L \sum_{p=\max(1,t-l)}^{t-1} \|\theta_p^\star - \theta_{p+1}^\star\|_2 + 2\beta_{t-1} \|A_t\|_{V_{t-1}^{-1}}.$$

By using the assumption $\forall a \in \mathcal{A}_t, |\langle A_t, \theta_t^\star \rangle| \leq 1$, we also have $r_t \leq 2$. So, with probability greater than $1 - \delta$,

$$r_t \leq 2L \sum_{p=\max(1,t-l)}^{t-1} \|\theta_p^\star - \theta_{p+1}^\star\|_2 + 2\beta_{t-1} \min\left(1, \|A_t\|_{V_{t-1}^{-1}}\right). \tag{15}$$

To conclude the proof, we use the results of Proposition 9.

Final step:

$$R_T = \sum_{t=1}^{T} r_t \leq 2L \sum_{t=1}^{T} \sum_{p=\max(1,t-l)}^{t-1} \|\theta_p^\star - \theta_{p+1}^\star\|_2 + 2\beta_T \sum_{t=1}^{T} \min\left(1, \|A_t\|_{V_{t-1}^{-1}}\right)$$

$$\leq 2L \sum_{t=1}^{T} \sum_{p=\max(1,t-l)}^{t-1} \|\theta_p^\star - \theta_{p+1}^\star\|_2 + 2\beta_T \sqrt{T} \sqrt{\sum_{t=1}^{T} \min\left(1, \|A_t\|_{V_{t-1}^{-1}}^2\right)}$$

$$\leq 2LB_T l + 2\sqrt{2}\beta_T \sqrt{dT} \sqrt{\lceil T/l \rceil} \sqrt{\log\left(1 + \frac{lL^2}{\lambda d}\right)}.$$

In the first inequality, we use the fact that $t \mapsto \beta_t$ is increasing. The second inequality is an application of the Cauchy-Schwarz inequality to the second term. The last inequality is an application of Proposition 9 □

By denoting $\tilde{O}$ the function growth when omitting the logarithmic terms, we have the following Corollary.

**Corollary 5** (Asymptotic regret bound for SW-LinUCB)**.** *If $B_T$ is known, by choosing $l = (\frac{dT}{B_T})^{2/3}$, the regret of the* SW-LinUCB *algorithm is asymptotically upper bounded with high probability by a term $\tilde{O}(d^{2/3}B_T^{1/3}T^{2/3})$ when $T \to \infty$.*

*If $B_T$ is unknown, by choosing $l = d^{2/3}T^{2/3}$, the regret of the* SW-LinUCB *algorithm is asymptotically upper bounded with high probability by a term $\tilde{O}(d^{2/3}B_T T^{2/3})$ when $T \to \infty$.*

*Proof.* With this particular choice of $l$, we have:

$$lB_T \sim d^{2/3}T^{2/3}B_T^{1/3}.$$

$\beta_T$ as defined by equation (13) is equivalent to $\sqrt{d\log(T)}$.

$\sqrt{T}\sqrt{\lceil T/l \rceil}$ has a similar behavior than $d^{-1/3}T^{1-1/3}B_T^{1/3}$, consequently the behavior of $\beta_T\sqrt{dT}\sqrt{\lceil T/l \rceil}\sqrt{\log\left(1 + \frac{lL^2}{\lambda d}\right)}$ is similar to $d^{2/3}B_T^{1/3}T^{2/3}\sqrt{\log(T)}\sqrt{\log(T/B_T)}$.

By neglecting the logarithmic term, we have with high probability,

$$R_T = \tilde{O}_{T\to\infty}(d^{2/3}B_T^{1/3}T^{2/3}).$$

$\square$