[Reviews · NeurIPS 2019]

Reviewer 1



Significance: the th1 can be useful in other works. The B_T term in Th2 (and the need of its knowledge to get the optimal \gamma) are little bit sad and could be more discussed (I'im wondering if on could get rid of it if the opponent had no knowledge of \theta_t but only \theta_{t-1} at timestep t) Originality: Results are very similar to SW-LinUCB since this work is essentially a replacement of the sliding window by a discounted weighted estimation. Quality: my main objection is about the adequacy to scenarios from real world (sec 4.2 is not very good). Anyway this is not the core of the paper since the proposed approach is likely to have the same kind of performances than any sliding windows technique. Clarity: the paper is cristal clear

Reviewer 2



Thanks for the author feedback. I am happy that the gap in regret bound is closed. However, it seems like the issue of parameter-free algorithms are not addressed. References [12,13] uses the SW-UCB as sub-routine to derive parameter-free bandit algorithms, it is extremely interesting to see what would happen if D-LinUCB is employed in such manner. -----------------------------------------------Original Comments---------------------------------- The paper provides an intuitive analysis for the D-LinUCB algorithm, which can attain a dynamic regret bound of order O(dB_T^{1/3}T^{2/3}). It also delivers solid experimental results on the proposed algorithm. The paper is also well written in general. But some questions remain: 1. It seems like the current algorithm is only optimal w.r.t. B_T and T, but not d, the problem dimension. It is proved in [5] and [12] that the lower bounds for K-armed and linear settings are O(K^{1/3}B_T^{1/3}T^{2/3}) and O(d^{2/3}B_T^{1/3}T^{2/3}), respectively. So it is worth checking the source of this gap. 2. It is shown in [12] that one could build a parameter-free algorithm for non-stationary linear bandit setting on top of the SW-LinUCB. It is thus expected that the D-LinUCB algorithm could also be enhanced accordingly. Is there any specific reason that hinders the flow? If not, what would be the parameter-free dynamic regret based on D-LinUCB? 3. It seems like the discounted linear regression estimator is not entirely new. In [1] B. Keskin and A. Zeevi. Chasing demand: Learning and Earning in a Changing Environment. In Mathematics of Operations Research, 42(2), 277–307, 2016. Similar type of estimator is analyzed for the 2-dimensional dynamic pricing setting. It is thus worth doing a thorough literature search to make sure all prior related works are properly credited. 4. At the end of paragraph 2 of Section 1.2, the dynamic regret bound is O(d^{2/3}B_T^{1/3}T^{2/3}).

Reviewer 3



Update (after reading the rebuttals): After reading the rebuttal of authors, I have addressed my concerns on the novelty of the new self-normalized concentration, since the key point is that the coefficient of regularizer is changing. I indeed appreciate this work. The idea of this paper is natural but there indeed exist technical challenges, and the authors address these issues elegantly. So I think it deserves an acceptance. Nevertheless, there are still many typos in current verison besides those listed before, for example, in Theorem 2, eq. (6), it should be "\log(1/\delta)" instead of "\log(1/\gamma)". I hope authors could carefully check and revise the final version if accepted :) ----------------------------------------------------------------- Original Comments: This paper extends the paper of [12,13] from a sliding-window strategy to the weighted least square method. The idea is natural and well-motivated. The authors give theoretical analysis, which is based on the self-normalized concentration from [1] with a clever choice of the norm. The analysis is essentially not hard but requires a deep understanding. Particularly, authors propose to consider the $M$-norm, where $M = V\tilde{V}^{-1}V$ in the WLS based linear bandits. This is by contrast with the common $V$-norm that considering in [1] and [12,13]. By considering the concentration over such a norm, authors can still appeal to the self-normalized concentration developed in [1] (with certain modifications) to WLS based linear bandits. Along with some standard techniques and tricks, they finish the argument. I appreciate the elegant idea. However, there are also some deficiencies in the current version, particularly regarding the paper organization. First, I would like to remark that the new self-normalized concentration claimed in the paper is essentially a direct extension of Theorem 1 of [1]. Since Proposition 1 is in the norm of \tilde{V} but not the $V$-norm, so authors can directly obtain the result from Theorem 1 of [1]. It seems unnecessary to restate the supermartingale argument as done in Lemma 1-3. Please clearly state the novelty of this part. Technical Issues: First, the current order for WLS bandits is O(dB_T^{1/3}T^{2/3}), the dependence in d is worse than that of sliding window linear bandits [12]. This can be improved by choosing \gamma as 1-(B_T/Td^{2/3})^{2/3} in line 470. line 461: the inequality should hold for the maximum singular value of $M$. How can you guarantee it is the same as the maximum eigenvalue, as $M$ may not be symmetric. Other minor issues: - line 459: there is a $\lambda$-term in $V$-martix, so is there an extra $\lambda$ missing in the 4-th inequality? - line 483: I do not think it is problematic to apply Theorem 2 of [12] to A_t, because the self-normalized concentration applies for all t>0. Authors are encouraged to add more explainations. Additionally, the related work in Section 1.2 is not satisfied. Authors are requested to reorganize the related work to make it more informative. - line 67-69, authors first introduce the full-information dynamic regret [6], but it follows by work of linear bandit with static regret [21,32]. The organization should be reconsidered. - line 76-77, authors claim "[2] gives fully problem-dependent dynamic regret bounds in O(log(T)).", however, it seems impossible to obtain a sublinear dynamic regret independent of the non-stationarity measure; besides, how can $O(log T)$ bound be regarded as "problem-dependent"? - line 78, the rate of [12,13] is not O(B_T^{2/3}T^{1/3}), the correct one should be O(B_T^{1/3}T^{2/3}), authors should carefully check it - line 83-84, I do not think [15] is the first work that considers the switching bandits. The seminal work of [Auer JMLR'02] has already given some results. [Auer JMLR'02] Auer, Peter. "Using confidence bounds for exploitation-exploration trade-offs." Journal of Machine Learning Research 3.Nov (2002): 397-422. Other related works on dynamic regret of contextual bandits are missing [Luo et al. COLT'18] and [Chen et al. COLT'19]. [Luo et al. COLT'18] Haipeng Luo, Chen-Yu Wei, Alekh Agarwal, and John Langford. Efficient contextual bandits in non-stationary worlds. In 31st Annual Conference on Learning Theory (COLT), 2018. [Chen et al. COLT'19] Yifang Chen, Chung-Wei Lee, Haipeng Luo, and Chen-Yu Wei. A New Algorithm for Non-stationary Contextual Bandits: Efficient, Optimal, and Parameter-free. In 32rd Annual Conference on Learning Theory (COLT), 2019. Some of the experimental descriptions are unclear and hard to understand. For example, line 280-282, what doe this sentence mean? - how to add the noise? - how to calculate the accuracy of the algorithm?

[Author Response · NeurIPS 2019]

We thank the reviewers for their feedback and detailed comments on the manuscript. We address below the most
important issues that will be fixed in the final version of the paper.

**Dependence in the dimension** $d$    Both Reviewers #2 and #3 noted that the dependence in $d$ would be expected to be of
the order of $d^{2/3}$, in light of [12]. This is indeed a relevant remark, and we confirm that by taking $\gamma = 1 - (B_T/(dT))^{2/3}$
(as proposed by Reviewer #3) one obtains a regret bound of this order. Regarding the comparison with [5] (Reviewer
#2): [5] relates to the non-contextual case, where the regret scales as $K^{1/3}$ only. This can be seen as a special case of
our model by choosing $K = d$ with fixed orthogonal contexts. However, in the general contextual case, the $d^{2/3}$ rate
appears as a consequence of the need to control deviations in all directions thus adding an additional $\sqrt{d}$ term in the
control of the stochastic term. Consequently, the optimal exponent of the dimension term to achieve the equilibrium
between the bias term and the stochastic term for the dynamic regret is $1/3$ in the case of a control for fixed directions
as in [5], but it is $2/3$ when the control is in all directions as in [12] and in our analysis.

**Novelty of the deviation bound**    We agree with Reviewer #3 that part of the arguments used in the proof of our
Theorem 1 are common with those introduced in [1]. However, Theorem 1 is not a mere consequence of the result
in [1]. To be more precise: Lemma 1 may indeed be seen as a simple extension of Lemma 8 of [1] to the case of
heteroscedastic noises (considering that the noise terms are given by $w_s\eta_s$). The proof of this lemma was included
to ease the understanding of the rest of the proof and to make the paper self-contained. Lemma 2 and 3 however
differ from Lemma 9 of [1] by the fact that we consider time-dependent regularization parameters ($\lambda_t$). As explained
in the main text, this is unavoidable when using exponential weights to avoid vanishing effect of the regularization.
Technically this implies that $\widetilde{M}_t$ defined in Lemma 2 is no more a supermartingale, although it still holds true that
$\mathbb{E}[\widetilde{M}_t] \leq 1$. In the weighted case, we also need to consider mixing constants ($\mu_t$) in the method of mixtures that differ
from the regularization parameters $\lambda_t$ (see Section 2). We will be more explicit regarding these differences with [1].

**Related works**    We thank Reviewers #2 and #3 for their comments that have improved greatly our literature review;
we have done our best to include all mentioned references. [Keskin & Zeevi], pointed by Reviewer #2 is an interesting
application-oriented paper on non-stationary bandits that has similarities with ours in the particular case where the
dimension $d = 2$, where they use a weighted estimator to construct first order optimal policies. It is true that the seminal
paper [Auer, 02] already considers shifting environments but they bound a different regret than the dynamic regret in
[15]; we'll comment on that. [Luo et al. COLT'18] and [Chen et al. COLT'19] will be discussed in the final version.
We also thank Reviewer #2 for correcting the typo in the order of the dynamic regret bound for [12,13] in line 78, where
exponents had been exchanged.

**Experimental section**    The objective of Section 4.2 was not to address a real-world application but rather to illustrate
the behavior of the algorithms in higher-dimensional problems. To do so, we have used the Criteo dataset only to
provide plausible context vectors and to avoid the peculiarities that would have arisen by, for instance, simulating these
at random on the hypercube as done in the small-dimensional example of Section 4.1. This experiment is interesting
in particular to confirm that the scaling with respect to the various parameters is correct, even if it does not solve a
real practical problem. This will be clarified in the text. In addition, we agree that the writing of this section could be
improved. We have simplified the preprocessing of the data and also incorporated the correct dependency on $d$ for $\gamma$
(see above), and the corresponding figure will be included in the final version of the paper.

We provide below answers regarding comments that do not necessitate significant modification of the paper.

- We agree with reviewers #1 and #2 that dependence on $B_T$ is currently a limitation of the proposed approach.
As pointed out by Reviewer #2, the method proposed in [12] could also be used with D-LinUCB. We are not
sure however that this approach would be practically satisfying for common values of the dimension $d$ and the
time-horizon $T$.
- (Reviewer #3) At line 461, the matrix $M$ refers to $V_{t-1}^{-1} \sum_{s=t-D}^{p} \gamma^{-s} A_s A_s^\top$ and hence is indeed a symmetric
matrix.
- (Reviewer #3) At line 483, the problem is the probabilistic nature of the statement: $|x^\top(\hat{\theta}_t - \theta_t)| \leq L$ for
all $x \in \mathbb{R}^d$ would indeed imply that $|X^\top(\hat{\theta}_t - \theta_t)| \leq L$ for any $\mathbb{R}^d$-valued random variable $X$. However,
$\mathbb{P}(|x^\top(\hat{\theta}_t - \theta_t)| > L) \leq \delta$ does not imply that $\mathbb{P}(|X^\top(\hat{\theta}_t - \theta_t)| > L) \leq \delta$, this can even be obviously wrong,
for instance, when $X = \hat{\theta}_t - \theta_t$. This gap justifies the inclusion of a corrected proof in Appendix C.

[Meta-Review · NeurIPS 2019]

Reviewers agree the paper is well-written and argued (aside from some typos which should be corrected). There is some question of novelty in light of building on techniques from previous work, but the algorithm's analysis and empirical performance place this paper above bar.